# Rotating 5D black holes:
# Interactions and deformations near extremality

**Alejandra Castro[1*], Juan F. Pedraza[2,3†], Chiara Toldo[1‡] and Evita Verheijden[1◦]**

**1** Institute for Theoretical Physics, University of Amsterdam,
Science Park 904, Postbus 94485, 1090 GL Amsterdam, The Netherlands
**2** Department of Physics and Astronomy, University College London, London WC1E 6BT, UK
**3** Martin Fisher School of Physics, Brandeis University, Waltham MA 02453, USA

⋆ a.castro@uva.nl, † j.pedraza@ucl.ac.uk, ‡ c.toldo@uva.nl, ◦ e.m.h.verheijden@uva.nl

## Abstract

We study a two-dimensional theory of gravity coupled to matter that is relevant to describe holographic properties of black holes with two equal angular momenta in five dimensions (with or without cosmological constant). We focus on the near-horizon geometry of the near-extremal black hole, where the effective theory reduces to Jackiw-Teitelboim (JT) gravity coupled to a massive scalar field. We compute the corrections to correlation functions due to cubic interactions present in this theory. A novel feature is that these corrections do not have a definite sign: for AdS$_5$ black holes the sign depends on the mass of the extremal solution. We discuss possible interpretations of these corrections from a gravitational and holographic perspective. We also quantify the imprint of the JT sector on the UV region, i.e. how these degrees of freedom, characteristic for the near-horizon region, influence the asymptotically far region of the black hole. This gives an interesting insight on how to interpret the IR modes in the context of their UV completion, which depends on the environment that contains the black hole.



# 1 Introduction

Despite the challenges of $AdS_2$ quantum gravity, in recent years nearly-$AdS_2$ spacetimes and their holographic properties have been a fertile setting to study semi-classical properties of black holes near extremality. $AdS_2$ is infamous for being difficult and confusing in comparison to its higher-dimensional cousins due to a variety of reasons [1–3]. However, it was observed in [4] that it was sensible and manageable to include the leading corrections away from pure $AdS_2$: this defines the concept of nearly-$AdS_2$ holography.

One of the defining properties of nearly-$AdS_2$ is the symmetry breaking pattern, which is universally encoded in two-dimensional Jackiw-Teitelboim gravity [5,6]. This gravitational theory contains a dilaton and the two-dimensional metric, and it perfectly captures the leading gravitational backreaction of $AdS_2$ [7]. It has also drawn a considerable amount of interest that the same symmetry breaking patterns are present in certain 1D quantum systems, such as the SYK model [8–14], which unveils interesting aspects of the holographic correspondence. We refer to [15] for an introductory overview of these developments, and [16] for a review on how these two-dimensional models serve as a platform to understand evaporating black holes.

Our intention here is to actually focus on aspects of nearly-$AdS_2$ that are less universal, and hence sensitive to the specificity of the theory and black hole. Although all extremal and near-extremal black holes share an $AdS_2$ factor in their near-horizon geometry, their backreaction could have additional features that need to be added to the simplest JT model (such as more degrees of freedom, interactions, and possibly a modification of boundary conditions). This has been most manifest in the context of rotating black holes in four and higher dimensions, and prior work that incorporates rotation in the holographic interpretation of nearly-$AdS_2$

includes [17–23].[1] In this work, our aim is to quantify how these additional features could affect the holographic dual of the nearly-AdS$_2$ region. We want to illustrate how the theory, e.g. if the black hole is embedded in AdS$_D$ or flat space, influences the interplay of JT with the whole black hole geometry. We also want to show that interactions among the JT sector and matter fields are not so straightforward to account for in the simplest SYK-like models.

Five-dimensional rotating black holes encompass a rich arena to explore these features. Here we will study a particular class of neutral black holes that have equal angular momenta, following the work in [18]. The appeal of this specific rotating black hole is that one can carry out explicitly a dimensional reduction to two dimensions, and hence place any analysis in clear resemblance or contrast with the predictions of JT gravity. As shown in [18], several aspects of the effective description of nearly-AdS$_2$ comply with JT gravity: there is a dilaton field $\mathcal{Y}$, whose operator dual has conformal dimension $\Delta_\mathcal{Y} = 2$. This already leads to important aspects of the effective boundary theory via a Schwarzian action, and a potential connection to SYK-like models. In addition, due to the rotation of the black hole, there is also a "squashing" mode $\mathcal{X}$ which is more irrelevant than the dilaton, $\Delta_\mathcal{X} > \Delta_\mathcal{Y}$. These two modes generally couple and must be considered together. An important part of our new contributions here is to quantify how these interactions alter the correlation functions in the nearly-AdS$_2$ region. This will provide crucial information that enters in the process of designing a dual description of these black holes that incorporates these new features.

Another important aspect we will explore is the relation between the five-dimensional theory, which provides the environment surrounding the black hole, with the nearly-AdS$_2$ description we described above. Our two-dimensional model encodes the entire environment of the black hole, interpolating between the near-horizon AdS$_2$ region and an asymptotically AdS$_5$ (or Minkowski$_5$) far region;[2] these are the two regions that we colloquially refer to as IR (AdS$_2$) and UV (AdS$_5$/Minkowski$_5$). This setup, in principle, gives a UV completion of the nearly-AdS$_2$ analysis that is tractable. In particular, we explore how the JT sector, where $\mathcal{Y}$ plays the prominent role, arises from the linear perturbations of the UV region. This requires a careful treatment of how the solutions behave as we take the decoupling limit that relates IR and UV regions, which we discuss in detail. We also study how the JT sector behaves in the far region, and highlight the sharp differences in the gluing procedure depending on the presence or absence of a negative cosmological constant. The way we set up the analysis of the decoupling limit is along the lines of [23] for the four-dimensional extreme Kerr solution. With regard to the AdS$_5$ aspects of our computations, a useful comparison is [30], where boundary gravitons of AdS$_3$ are interpolated to AdS$_5$.

This paper is organized as follows. We start in Sec. 2 by reviewing the two-dimensional theory that one obtains by dimensional reduction of the five-dimensional Einstein-Hilbert action with a negative cosmological constant. In Sec. 3 we initiate our analysis of the effective field theory description of the IR theory for this two-dimensional system. Here we carefully quantify how the JT sector interacts with the massive mode $\mathcal{X}$, and quantify the role of these interactions in the two- and four-point functions involving $\mathcal{X}$. We find that cubic couplings among $\mathcal{X}$ and $\mathcal{Y}$ do not have a definite sign: the sign depends on the extremal values that define the AdS$_2$ background. This introduces an interesting feature when attempting to reproduce such an effect in a holographic dual. In Sec. 4 we discuss how to single out the JT sector from the gravitational perturbations in the black hole background. We carefully discuss the decoupling limit of the UV perturbations, and how to extrapolate these perturbations to the UV boundary. We end in Sec. 5 with a careful discussion of our results. In particular, we

---

[1]The three-dimensional BTZ black hole is another example of a rotating black hole, and many aspects are well described within JT gravity. See [24–29] for extensive work on this direction.

[2]Here we focus on the case of negative or zero cosmological constant; it would be interesting to consider $\Lambda > 0$ and carry out the analysis in the context of 5D de Sitter black holes. We leave this as a future direction.

embed our results in the context of other well known gravitational properties of AdS$_5$ black holes that would be interesting to incorporate in future aspects of nearly-AdS$_2$ holography; and we discuss how to account in a dual description for the effects of the interactions among $\mathcal{Y}$ and $\mathcal{X}$. We also included two appendices: in App. A we review some aspects of the AdS$_5$ black holes studied here, with emphasis on the near-extremal limit; and in App. B we collect the technical aspects of computing Witten diagrams in the context of nearly-AdS$_2$ spacetimes.

## 2 Dimensional reduction from 5D to 2D

Following [18], our starting point is the five-dimensional Einstein-Hilbert action with a negative cosmological constant,

$$I_{5D} = \frac{1}{2\kappa_5^2} \int d^5 x \sqrt{-g^{(5)}} \left( \mathcal{R}^{(5)} + \frac{12}{\ell_5^2} \right) . \tag{1}$$

From here we will construct a two-dimensional effective theory by performing a dimensional reduction. The decomposition of the five-dimensional metric we will consider is

$$g_{\mu\nu}^{(5)} dx^\mu dx^\nu = e^{\psi+\chi} ds_{(2)}^2 + R^2 e^{-2\psi+\chi} d\Omega_2^2 + R^2 e^{-2\chi} \left( \sigma^3 + A \right)^2 , \tag{2}$$

where we introduced a two-dimensional metric

$$ds_{(2)}^2 = g_{ab} dx^a dx^b , \qquad a, b = 0, 1 , \tag{3}$$

two scalar fields, $\psi$ and $\chi$, and a one form $A = A_a dx^a$. The scale $R$ is introduced to keep the scalar fields dimensionless. Here, the fields $(g_{ab}, A_a, \psi, \chi)$ depend on the two-dimensional variables $x^a$, i.e. they are the 'massless' degrees of freedom in the context of the dimensional reduction. These backgrounds exhibit a manifest two-sphere

$$d\Omega_2^2 = d\theta^2 + \sin^2\theta d\phi^2 = (\sigma^1)^2 + (\sigma^2)^2 , \tag{4}$$

where the angular forms are

$$\begin{aligned}
\sigma^1 &= -\sin\hat{\psi} d\theta + \cos\hat{\psi}\sin\theta d\phi , \\
\sigma^2 &= \cos\hat{\psi} d\theta + \sin\hat{\psi}\sin\theta d\phi , \\
\sigma^3 &= d\hat{\psi} + \cos\theta d\phi .
\end{aligned} \tag{5}$$

In total, the decomposition (2) has $SU(2)\times U(1)$ symmetry. The ansatz (2) will accommodate, among other solutions, black holes in AdS$_5$ with equal angular momenta [31]. These black hole solutions will be described in detail in Sec. 4. In Sec. 2.1 we briefly discuss the Myers-Perry configuration [32] as a concrete example for the case $\ell_5 \to \infty$.

In terms of the two-dimensional fields, the dimensional reduction of (1) reads [18]

$$\begin{aligned}
I_{2D} = \frac{1}{2\kappa_2^2} \int d^2 x \sqrt{-g}\, e^{-2\psi} \Big( \mathcal{R} - \frac{R^2}{4} e^{-3\chi-\psi} F^2 - \frac{3}{2}(\nabla\chi)^2 \\
+ \frac{1}{2R^2}\left( 4e^{3\psi} - e^{5\psi-3\chi} \right) + \frac{12}{\ell_5^2} e^{\psi+\chi} \Big) ,
\end{aligned} \tag{6}$$

where $\kappa_2^2 = \frac{\kappa_5^2}{16\pi^2 R^3}$, $\mathcal{R}$ is the Ricci scalar associated to (3), and we introduced the field strength $F = dA = \partial_a A_b\, dx^a \wedge dx^b$. From $I_{2D}$ we can see more clearly the role that each scalar field has in this effective 2D description. The field $\psi$ plays the role of a dilaton: the fields were introduced

in (2) such that $\psi$ lacks a kinetic term in (6), and it controls the area of the squashed $S^3$ in (2). In contrast, the field $\chi$ couples more traditionally to the metric $g_{ab}$ and the remaining fields, and its non-trivial profile reflects that the $S^3$ is squashed.

The equations of motion for the two-dimensional fields are

$$e^{2\psi}(\nabla_a\nabla_b - g_{ab}\Box)e^{-2\psi} + g_{ab}\left(\frac{1}{4R^2}\left(4e^{3\psi} - e^{5\psi-3\chi}\right) + \frac{R^2}{8}e^{-3\chi-\psi}F^2 + \frac{6}{\ell_5^2}e^{\psi+\chi}\right)$$

$$+ \frac{3}{2}\left(\nabla_a\chi\nabla_b\chi - \frac{1}{2}g_{ab}(\nabla\chi)^2\right) = 0\ ,$$

$$\mathcal{R} + \frac{3}{4}e^{-3\chi+5\psi}\left(\frac{1}{R^2} - \frac{R^2}{2}F^2e^{-6\psi}\right) - \frac{1}{R^2}e^{3\psi} + \frac{6}{\ell_5^2}e^{\psi+\chi} - \frac{3}{2}(\nabla\chi)^2 = 0\ ,$$

$$e^{2\psi}\nabla_a(e^{-2\psi}\nabla^a\chi) + \frac{R^2}{4}e^{-3\chi-\psi}F^2 + \frac{1}{2R^2}e^{5\psi-3\chi} + \frac{4}{\ell_5^2}e^{\psi+\chi} = 0\ ,$$

$$\nabla_a\left(e^{-3\psi-3\chi}F^{ab}\right) = 0\ . \tag{7}$$

It is very important to stress that *all* solutions to the equations of motion derived from (6) are solutions to the 5D Einstein equations: this makes $I_{2D}$ a consistent truncation of $I_{5D}$ as shown in [18]. The last equation of motion in (7) is straightforward to solve and gives

$$F_{ab} = Qe^{3\psi+3\chi}\epsilon_{ab}\ , \qquad F^2 = -2Q^2e^{6\psi+6\chi}\ , \tag{8}$$

where $Q$ is a constant, and $\epsilon_{ab}$ is the two-dimensional Levi-Civita tensor. In the two-dimensional context, $Q$ is the electric charge; from a five-dimensional perspective it controls the angular momentum of (2). Using this solution, one can integrate out the field strength and the resulting action is

$$I_{2D} = \frac{1}{2\kappa_2^2}\int d^2x\sqrt{-g}\,e^{-2\psi}\Big(\mathcal{R} - \frac{R^2Q^2}{2}e^{3\chi+5\psi} - \frac{3}{2}(\nabla\chi)^2$$

$$+ \frac{1}{2R^2}\left(4e^{3\psi} - e^{5\psi-3\chi}\right) + \frac{12}{\ell_5^2}e^{\psi+\chi}\Big)\ . \tag{9}$$

We will be interested in studying the following two aspects of this theory, and the interplay between them.

**IR EFT:** One of the simplest solutions to $I_{2D}$ corresponds to the case when $\psi$ and $\chi$ are constants, which implies that $g_{ab}^{(2)}$ is locally AdS$_2$. This is what we denote as the IR solution, and from a five-dimensional perspective it corresponds to the near-horizon geometry of a rotating black hole in AdS$_5$/Minkowski$_5$. One can deform this background to allow for a nearly-AdS$_2$ geometry, as done in [18]; in holographic jargon, this corresponds to turning on an irrelevant deformation. Here we will study aspects of the effective field theory near the IR fixed point, with particular emphasis on quantifying the effects of interactions among the fields at leading order in the deformation.

**UV/IR connection:** There are other solutions of this theory where $\psi$ and $\chi$ have a non-trivial radial profile. We will be interested in setups that accommodate asymptotically AdS$_5$/Minkowski$_5$ solutions, and in particular the rotating AdS$_5$ black holes. These will be denoted as the UV solutions. For these solutions, we want to carefully connect them to the EFT that describes the IR; our aim is to quantify the imprint of the nearly-AdS$_2$ deformations on the boundary of AdS$_5$ or Minkowski$_5$.

Each of these configurations will be constructed and described in subsequent sections. In Sec. 3 we will discuss the IR side, with emphasis on the interactions appearing in nearly-AdS$_2$ and its holographic interpretation. Sec. 4 is devoted to the UV/IR connection, where we will place the IR analysis in the context of asymptotically AdS$_5$ (or Minkowski$_5$) region.

## 2.1 Example: the Myers-Perry black hole

The Myers-Perry configuration [32], describing a five-dimensional neutral rotating black hole with two coincident angular momenta, is a concrete example of a five-dimensional solution that fits into the framework mentioned above when $\ell_5 \to \infty$. Following the notation we are using in (2), it is a stationary solution of the form

$$g_{\mu\nu}^{(5)}\mathrm{d}x^\mu\mathrm{d}x^\nu = e^{\psi+\chi}g_{ab}^{(2)}\mathrm{d}x^a\mathrm{d}x^b + R^2 e^{-2\psi+\chi}\,d\Omega_2^2 + R^2 e^{-2\chi}(\sigma^3+A)^2 \,, \tag{10}$$

where the 2D metric $g_{ab}^{(2)}$ is specified by

$$e^{\psi+\chi}g_{ab}^{(2)}\mathrm{d}x^a\mathrm{d}x^b = \frac{\mathrm{d}\hat{r}^2}{\Delta(\hat{r})} - \Delta(\hat{r})e^{-2\psi+3\chi}\mathrm{d}\hat{t}^2 \,, \tag{11}$$

with

$$\Delta(\hat{r}) = 1 - \frac{2m}{\hat{r}^2} + \frac{2ma^2}{\hat{r}^4} \,. \tag{12}$$

The warp factors are

$$R^2 e^{-2\chi} = \frac{\hat{r}^2}{4} + \frac{ma^2}{2\hat{r}^2} \,, \qquad R^2 e^{-2\psi} = \frac{\hat{r}^2}{4}\,e^{-\chi} \,, \tag{13}$$

and the gauge field that captures the rotational nature of the solution is

$$A = \frac{a}{2R^2}\left(-\frac{2m}{\hat{r}^2}\right)e^{2\chi}\,\mathrm{d}\hat{t} \,. \tag{14}$$

This metric describes an asymptotically flat black hole; also note that it is the Kaluza-Klein black hole in [33]. In Sec. 4.1 we will consider a generalization of this metric to include the presence of a (negative) cosmological constant [31], and among other things we will discuss the near-horizon limit in both cases.

# 3 Effective field theory from the IR

This section is devoted to the analysis of the two-dimensional theory from the perspective of the IR. We will first review aspects of the IR fixed point and the linear analysis, which was done in [18], and then perform an analysis of the interactions among the fields around this fixed point.

## 3.1 Linearized equations of motion and the JT sector

The IR background corresponds to the solution to the equations of motion (7) with constant scalars:

$$e^{2\psi(x)} = e^{2\psi_0} \,, \qquad e^{2\chi(x)} = e^{2\chi_0} \,, \tag{15}$$

with $\psi_0$ and $\chi_0$ constants. The equations of motion then determine

$$\frac{R^4 Q^2}{2}e^{3\chi_0} = e^{-3\chi_0} - e^{-2\psi_0} \,, \qquad \frac{1}{\ell_5^2} = \frac{1}{8R^2}e^{4\psi_0-\chi_0}\left(e^{-3\chi_0} - 2e^{-2\psi_0}\right) \,, \tag{16}$$

where $Q$ is defined in (8), and they also imply that the Ricci scalar is given by

$$\mathcal{R} = -\frac{2}{\ell_2^2} \,, \qquad \frac{1}{\ell_2^2} = \frac{1}{2R^2}e^{3\psi_0}\left(-4 + 3e^{2\psi_0-3\chi_0}\right) \,, \tag{17}$$

indicating that the metric is locally AdS$_2$. As was done in [18], it will prove useful to trade $Q$ for the constant $q$, which is defined as

$$q \equiv \frac{1}{8}e^{2\psi_0}(R^4 Q^2 e^{3\chi_0} - e^{-3\chi_0}) = \frac{1}{8}\left(e^{2\psi_0 - 3\chi_0} - 2\right) . \tag{18}$$

In terms of $q$, we then find

$$\frac{1}{\ell_2^2} = \frac{1}{R^2}e^{3\psi_0}(1 + 12q) , \qquad \frac{1}{\ell_5^2} = \frac{q}{R^2}e^{2\psi_0 - \chi_0} . \tag{19}$$

These relations define the IR background solution. Note that the limit $q \to 0$ corresponds to setting the five-dimensional cosmological constant to zero, and $q \to \infty$ would be a very strongly coupled AdS$_5$ spacetime.

Next, we move on to characterize perturbations around the IR background, first focusing on the linear behavior of the perturbations. We define

$$
\begin{aligned}
\mathcal{Y} &\equiv e^{-2\psi} - e^{-2\psi_0} , \\
\mathcal{X} &\equiv \chi - \chi_0 , \\
h_{ab} &\equiv g_{ab} - \bar{g}_{ab} ,
\end{aligned}
\tag{20}
$$

where $\bar{g}_{ab}$ is a locally AdS$_2$ metric with AdS radius given by (17). The linearized equations of motion obtained from (7) become

$$\left(\bar{\nabla}_a \bar{\nabla}_b - \bar{g}_{ab}\bar{\Box}\right)\mathcal{Y} + \frac{1}{\ell_2^2}\bar{g}_{ab}\mathcal{Y} = 0 , \tag{21}$$

$$\bar{\Box}\mathcal{X} + \frac{2}{R^2}e^{3\psi_0}(1 - 2e^{-3\chi_0 + 2\psi_0})\mathcal{X} + \frac{1}{R^2}e^{5\psi_0}(-2 + e^{-3\chi_0 + 2\psi_0})\mathcal{Y} = 0 , \tag{22}$$

$$\delta\mathcal{R} - \frac{3}{R^2}\left(2e^{3\psi_0} - e^{-3\chi_0 + 5\psi_0}\right)\mathcal{X} + \frac{6}{R^2}e^{5\psi_0}(1 - e^{-3\chi_0 + 2\psi_0})\mathcal{Y} = 0 . \tag{23}$$

Here, the bar indicates that the covariant derivatives and Laplacians are with respect to $\bar{g}_{ab}$. We also have $\mathcal{R} = \bar{\mathcal{R}} + \delta\mathcal{R}$, where the linear response of the Ricci scalar is

$$\delta\mathcal{R} = -\bar{\mathcal{R}}^{ab}h_{ab} + \bar{\nabla}^a\bar{\nabla}^b h_{ab} - \bar{\Box}h^a_{\ a} , \tag{24}$$

and $\bar{\mathcal{R}}$ is given by (17).

It is clear that (21) is the equation of motion characteristic of JT gravity. However, as noted in [18], the new feature here is that $\mathcal{Y}$ is coupled to $\mathcal{X}$ via (22), and hence slightly more work is needed to single out the JT sector. The solutions to (22) split into a homogeneous and inhomogeneous part,

$$\mathcal{X} = \mathcal{X}_{\text{inh}} + \mathcal{X}_{\text{hom}} , \tag{25}$$

where the inhomogeneous solution is

$$\mathcal{X}_{\text{inh}} = \frac{2q}{1 + 2q}e^{2\psi_0}\mathcal{Y} , \tag{26}$$

and the homogeneous solution satisfies

$$\bar{\Box}\mathcal{X}_{\text{hom}} - \frac{2}{\ell_2^2}\frac{(3 + 16q)}{(1 + 12q)}\mathcal{X}_{\text{hom}} = 0 . \tag{27}$$

From here we see that $\mathcal{X}_{\text{hom}}$ is an irrelevant perturbation of the AdS$_2$ background, and the conformal dimension is given by

$$\Delta_{\mathcal{X}} = \frac{1}{2}\left(1 + 5\sqrt{\frac{1 + \frac{28}{5}q}{1 + 12q}}\right) . \tag{28}$$

For comparison, the field $\mathcal{Y}$ has $\Delta_{\mathcal{Y}} = 2$, and hence $\Delta_{\mathcal{Y}} < \Delta_{\mathcal{X}} \leq 3$.

There are also couplings to the metric perturbations via (23). To solve this equation, it is convenient to pick coordinates for the background metric and fix a gauge for $h_{ab}$. We will use Poincaré coordinates

$$\bar{g}_{ab}\mathrm{d}x^a\mathrm{d}x^b = \ell_2^2\left(-r^2\mathrm{d}t^2 + \frac{\mathrm{d}r^2}{r^2}\right),\tag{29}$$

and a convenient way to write the metric perturbations $h_{ab}$ is[3]

$$h_{ab} = \bar{g}_{ab}\hat{h}_{ab},\tag{30}$$

and hence $h_{tr} = 0$. Using this parametrization we find that (23) reduces to

$$\frac{1}{\ell_2^2}\left(\frac{1}{r^2}\partial_t^2\hat{h}_{rr} + r\partial_r\hat{h}_{rr} + 2\hat{h}_{rr}\right) - \frac{1}{\ell_2^2}\left(3r\partial_r\hat{h}_{tt} + r^2\partial_r^2\hat{h}_{tt}\right)$$
$$+ \frac{24}{R^2}e^{3\psi_0}q\,\mathcal{X} - \frac{6}{R^2}e^{5\psi_0}(1+8q)\mathcal{Y} = 0.\tag{31}$$

The solutions to (31) split into

$$\hat{h}_{tt} = \frac{6q}{1+2q}\mathcal{X} + \alpha_1\mathcal{Y} + \hat{h},$$
$$\hat{h}_{rr} = \frac{6q}{1+2q}\mathcal{X} + \alpha_1\mathcal{Y} + \hat{h}_{rr}^{(\mathrm{hom})},\tag{32}$$

where $\alpha_1$ is an arbitrary constant, and $\hat{h}_{rr}^{(\mathrm{hom})}$ is the homogeneous solution to the kinetic terms acting on $\hat{h}_{rr}$. The perturbation $\hat{h}$ satisfies

$$\left(3r\partial_r\hat{h} + r^2\partial_r^2\hat{h}\right) + 6e^{2\psi_0}\frac{1+10q+8q^2}{(1+2q)(1+12q)}\mathcal{Y} = 0.\tag{33}$$

This equation can be easily solved as a radial integral of $\mathcal{Y}$. In particular, for AdS$_2$ in Poincaré coordinates we find that the solutions to (21) are[4]

$$\mathcal{Y}(r,t) = c^-r + c^0rt + c^+(rt^2 - \frac{1}{r}),\tag{34}$$

with $c^i$ constants, and hence (33) gives

$$\hat{h}(r,t) = -2e^{2\psi_0}\frac{1+10q+8q^2}{(1+2q)(1+12q)}\left(c^+(\frac{3}{r} + rt^2) + c^-r + c_0tr\right) + \hat{h}^{(\mathrm{hom})},\tag{35}$$

where we included the homogeneous solution to (33). Note that the homogeneous solutions to the metric perturbations in two-dimensions are trivial and can be reabsorbed as part of the background solution; we are writing them here just as a formality.

**JT sector: Nearly-AdS$_2$ background.** We have completely solved the linear perturbations around the fixed point, and from here we want to single out the part of those perturbations that we will denote as the JT sector, also known as the nearly-AdS$_2$ background. These are the

---

[3]Indices are not summed over in (30); it is just to indicate that the perturbation is "proportional" to the background metric.

[4]The notation $c^i$ with $i = \pm, 0$ follows from the discussion in [23]. As done there, and originally in [7], one can relate each of these constants to the $sl(2)$ isometries of the AdS$_2$ background.

perturbations that mimic the effects of JT gravity, and hence are responsible for the leading effects in terms of the gravitational backreaction. In the notation used here, we see that for

$$\mathcal{X} = \frac{2q}{1+2q}e^{2\psi_0}\mathcal{Y}\,,\tag{36}$$

the dynamics reduce to the JT equations of motion

$$\left(\bar{\nabla}_a\bar{\nabla}_b - \bar{g}_{ab}\bar{\Box}\right)\mathcal{Y} + \frac{1}{\ell_2^2}\bar{g}_{ab}\mathcal{Y} = 0\,.\tag{37}$$

Then, the backreaction of the metric is just given by (33). In short, the JT sector corresponds to turning off all of the homogeneous solutions to $\mathcal{X}$ and $h_{ab}$ found above. For further aspects of the JT sector, and in particular for a discussion of how the Schwarzian effective action appears in this theory, we refer to [18].

## 3.2 Effective action and interactions

Next, we wish to study the IR effective action perturbatively around the background solution (15)-(17). To do so, we subtract the inhomogeneous pieces and redefine the perturbations as follows:[5]

$$\begin{aligned}
\psi &= \psi_0 - \epsilon\frac{e^{2\psi_0}}{2}\mathcal{Y}\,,\\
\chi &= \chi_0 + \epsilon\left(\frac{2q}{1+2q}e^{2\psi_0}\mathcal{Y} + \hat{\mathcal{X}}\right)\,,\\
g_{tt} &= \bar{g}_{tt} + \epsilon\left(\bar{g}_{tt}\frac{6q}{1+2q}\hat{\mathcal{X}} + h_{tt}\right)\,,\\
g_{rr} &= \bar{g}_{rr} + \epsilon\left(\bar{g}_{rr}\frac{6q}{1+2q}\hat{\mathcal{X}} + h_{rr}\right)\,.
\end{aligned}\tag{38}$$

We introduced a dimensionless parameter $\epsilon$ to keep track of the various orders in the perturbations, but we will set $\epsilon = 1$ at the end of the calculations. The decomposition above is catered to single out the JT sector $(\mathcal{Y}, h_{ab})$ from the massive degree of freedom $\hat{\mathcal{X}}$, so that they are decoupled at quadratic order as we will show below.

With the above definitions, we can now expand the action (6) at different orders in $\epsilon$, while keeping only terms linear in $\mathcal{Y}$. Basically, we want to capture the leading interactions of the JT sector with the field $\hat{\mathcal{X}}$. The resulting terms are:

- Linear order:
  At linear order we only find total derivatives; this shows consistency of the IR fixed point.

- Quadratic order:
  At quadratic order we find a free theory for $\hat{\mathcal{X}}$ (up to total derivatives), as expected:[6]

$$\mathcal{L}_2 = \frac{3e^{-2\psi_0}}{2\kappa_2^2}\left[-\frac{1}{2}(\bar{\nabla}\hat{\mathcal{X}})^2 - \frac{1}{2}m_\mathcal{X}^2\hat{\mathcal{X}}^2\right]\,,\tag{39}$$

  where

$$m_\mathcal{X}^2 \equiv \frac{1}{\ell_2^2}\frac{6+32q}{1+12q}\,.\tag{40}$$

---

[5]Note that the first equation is the linearized version of (20).
[6]We normalize $\mathcal{L}$ such that $\tilde{S}_{2D} = \int d^2x\sqrt{-\bar{g}}\,\mathcal{L}$.

All indices in the effective action are lowered and raised with the background metric $\bar{g}_{ab}$. From the above mass term we can read off the conformal dimension of the operator dual to $\mathcal{X}$, which coincides with (28):

$$\Delta_{\mathcal{X}} = \frac{1 + \sqrt{1 + 4m_{\mathcal{X}}^2 \ell_2^2}}{2} = \frac{1}{2}\left(1 + 5\sqrt{\frac{1 + \frac{28}{5}q}{1 + 12q}}\right). \tag{41}$$

- Cubic order:
  At cubic order we find the following interaction terms (up to total derivatives), which we regroup in three categories:

$$\mathcal{L}_3^{(a)} = \frac{3e^{-2\psi_0}}{2\kappa_2^2}\left[-\frac{10q(7 + 32q)}{3\ell_2^2(1 + 2q)(1 + 12q)}\hat{\mathcal{X}}^3 - \frac{12q^2 e^{2\psi_0}}{(1 + 2q)^2}\hat{\mathcal{X}}(\bar{\nabla}\hat{\mathcal{X}})(\bar{\nabla}\mathcal{Y})\right.$$
$$\left.-\frac{e^{2\psi_0}}{2}\mathcal{Y}(\bar{\nabla}\hat{\mathcal{X}})^2 + \frac{e^{2\psi_0}}{2\ell_2^2}\frac{(1 + 6q)(9 + 38q - 80q^2)}{(1 + 2q)^2(1 + 12q)}\mathcal{Y}\hat{\mathcal{X}}^2\right], \tag{42}$$

$$\mathcal{L}_3^{(b)} = \frac{3e^{-2\psi_0}}{2\kappa_2^2}\left[-\frac{1}{2\ell_2^2}\frac{(3 + 16q)}{(1 + 12q)}h^a{}_a\hat{\mathcal{X}}^2 - \frac{1}{4}h^a{}_a(\bar{\nabla}\hat{\mathcal{X}})^2 + \frac{1}{2}h^{ab}\bar{\nabla}_a\hat{\mathcal{X}}\bar{\nabla}_b\hat{\mathcal{X}}\right], \tag{43}$$

$$\mathcal{L}_3^{(c)} = \frac{1}{8\kappa_2^2}\left[2\mathcal{Y}h_{ab}\bar{\nabla}^a\bar{\nabla}^b h^c{}_c - 4\mathcal{Y}h_a{}^b\bar{\nabla}^a\bar{\nabla}^c h_{bc} + 2\mathcal{Y}h^{ab}\bar{\Box}h_{ab}\right.$$
$$\left.-2\mathcal{Y}(\bar{\nabla}^a h_{ab})(\bar{\nabla}_c h^{bc}) + \mathcal{Y}(\bar{\nabla}^a h^b{}_b)(\bar{\nabla}_a h^c{}_c) + \mathcal{Y}(\bar{\nabla}^a h_{bc})(\bar{\nabla}_a h^{bc})\right]. \tag{44}$$

The first type of interactions, $\mathcal{L}_3^{(a)}$, involve $\hat{\mathcal{X}}$ and $\mathcal{Y}$ only. Here $\hat{\mathcal{X}}$ is the propagating degree of freedom, while $\mathcal{Y}$ can be considered as a background field (i.e. the field that defines the nearly-AdS$_2$ background). The second type, $\mathcal{L}_3^{(b)}$, contains $\mathcal{O}(h)$ terms, so they give a three-point vertex involving one graviton leg. Finally, the third type contains $\mathcal{O}(h^2)$ terms, which can be interpreted as a kinetic term for the graviton. We emphasize that $\mathcal{O}(h^2)$ terms cannot appear on their own, because gravity in two dimensions is topological. They appear in our system because the graviton couples to $\mathcal{Y}$.

We could continue to higher orders in the expansion; however, the quadratic and cubic order terms will be enough for the purposes of our calculations.

## 3.3 Graviton propagator

The graviton propagator can be obtained from the terms denoted as $\mathcal{L}_3^{(c)}$, which will initiate our discussion on how to normalize our fields and compare cubic terms. To simplify the calculation, it will be convenient to pick a suitable gauge; we will use (29) for the background metric. Following [24], we thus define

$$h_{tt} = -\ell_2^2 r^2 (h + g), \qquad h_{rr} = \frac{\ell_2^2}{r^2}(h - g). \tag{45}$$

Implementing these replacements, and integrating by parts several times, we can massage the terms in $\mathcal{L}_3^{(c)}$ into the following form:

$$\mathcal{L}_3^{(c)} = \frac{3e^{-2\psi_0}}{2\kappa_2^2}\left[h^2(\cdots) + h(\cdots) + (h\text{-independent terms})\right]. \tag{46}$$

Here, the $(\cdots)$ are terms independent of $h$, reflecting the fact that $h$ is not a dynamical field. Varying with respect to $h$ we can then derive an algebraic equation of motion for $h$, which can be solved to obtain

$$h = \frac{1}{2r^2\mathcal{Y}}\left[r^3(2g + rg')\mathcal{Y}' + \dot{g}\,\ddot{\mathcal{Y}}\right]. \tag{47}$$

As in [24], we focus on the time-independent solution $\mathcal{Y} = a\,r$, where $a$ is the (dimensionful) parameter that sources the irrelevant deformation. In this case we find:

$$h = g + \frac{1}{2}rg'. \tag{48}$$

Plugging this back into (46) and integrating by parts again we obtain

$$\mathcal{L}_3^{(c)} = -\frac{a\,r^3}{8\kappa_2^2\ell_2^2}g'^2. \tag{49}$$

This is exactly the same kinetic term that was found in [24] for pure JT gravity. Notice that in their coordinates $r \to 1/z$ (and hence $\partial_r \to -z^2\partial_z$) we get

$$\mathcal{L}_3^{(c)} = -\frac{a\,z}{8\kappa_2^2\ell_2^2}(\partial_z g)^2. \tag{50}$$

Since $\sqrt{-\bar{g}} = \ell_2^2/z^2$ in these coordinates and $\kappa_2^2 = 8\pi G_N$, we recover the first term in their equation (3.8):

$$S_{2D}^g = \int dt\,dz\,\sqrt{-g}\,\mathcal{L}_3^{(c)} = \int dt\,dz\left(-\frac{a}{64\pi G_N z}(\partial_z g)^2\right). \tag{51}$$

This leads to the following (bulk-to-bulk) propagator for the graviton:

$$G_g(t, z; t', z') = -\frac{i16\pi G_N}{a}\left[z^2\Theta(z' - z) + z'^2\Theta(z - z')\right]\delta(t - t'). \tag{52}$$

The absence of time derivatives in the kinetic term implies that the propagator is instantaneous in time. This means that, despite appearances, $g$ is not a propagating degree of freedom. However, when coupled to matter fields, the above propagator mediates their backreaction.

## 3.4 Field rescaling and diagram analysis

The graviton propagator is proportional to $G_N$ (or $\kappa_2^2$), so we can treat the interactions with the graviton (backreaction) perturbatively. To do so, we rescale the scalar field $\hat{\mathcal{X}}$ as

$$\hat{\mathcal{X}} \to \sqrt{\frac{2\kappa_2^2}{3e^{-2\psi_0}}}\check{\mathcal{X}}, \tag{53}$$

so that at quadratic order

$$\mathcal{L}_2 = -\frac{1}{2}(\bar{\nabla}\check{\mathcal{X}})^2 - \frac{1}{2}m_{\mathcal{X}}^2\check{\mathcal{X}}^2, \tag{54}$$

at cubic order

$$\begin{aligned}
\mathcal{L}_3^{(a)} = &-\sqrt{\frac{2\kappa_2^2}{3e^{-2\psi_0}}}\frac{10q(7 + 32q)}{3\ell_2^2(1 + 2q)(1 + 12q)}\check{\mathcal{X}}^3 - \frac{12q^2 e^{2\psi_0}}{(1 + 2q)^2}\check{\mathcal{X}}(\bar{\nabla}\check{\mathcal{X}})(\bar{\nabla}\mathcal{Y}) \\
&- \frac{e^{2\psi_0}}{2}\mathcal{Y}(\bar{\nabla}\check{\mathcal{X}})^2 + \frac{e^{2\psi_0}}{2\ell_2^2}\frac{(1 + 6q)(9 + 38q - 80q^2)}{(1 + 2q)^2(1 + 12q)}\mathcal{Y}\check{\mathcal{X}}^2,
\end{aligned} \tag{55}$$

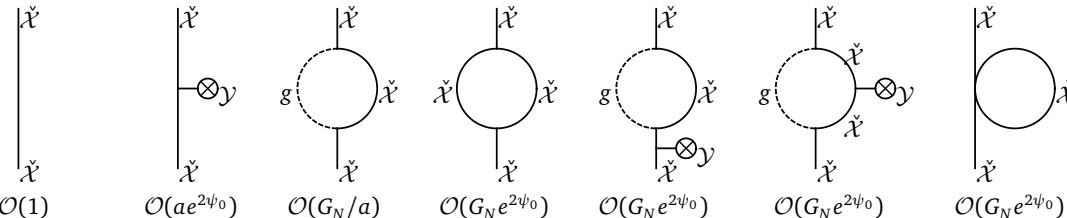

Figure 1: Leading diagrams (up to one loop) contributing to the two-point function, by order of relevance. Notice that the last four loop diagrams contribute at the same order, and we have included the loop correction due to a quartic interaction.

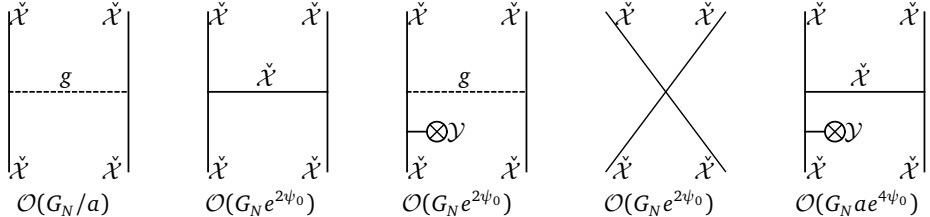

Figure 2: Leading (tree level) diagrams contributing to the four-point function. The second, third and fourth diagrams enter at the same order, and they give the next leading correction to the gravitational backreaction. Notice that the fourth diagram is due to a quartic coupling.

and

$$\mathcal{L}_3^{(b)} = -\frac{1}{2\ell_2^2}\frac{(3+16q)}{(1+12q)}h^a{}_a\check{\mathcal{X}}^2 - \frac{1}{4}h^a{}_a(\bar{\nabla}\check{\mathcal{X}})^2 + \frac{1}{2}h^{ab}\bar{\nabla}_a\check{\mathcal{X}}\bar{\nabla}_b\check{\mathcal{X}}. \qquad (56)$$

The first term in $\mathcal{L}_3^{(a)}$ (i.e. the cubic term in $\check{\mathcal{X}}$) is suppressed by a factor of $\kappa_2 \propto G_N^{1/2}$. This term will give the leading behavior of the three-point function. It will also correct the two-point function via loop diagrams (see Fig. 1). Similarly, it will also contribute to the four-point function via an exchange diagram, but this will be subleading with respect to the graviton exchange (see Fig. 2). The remaining terms in $\mathcal{L}_3^{(a)}$ will give a correction to the two-point function, with $\mathcal{Y}$ treated as a background field. These corrections will be suppressed by a factor of $\mathcal{O}(ae^{2\psi_0}) \ll 1$.[7] However, they will be more important than any of the loop diagrams, because these will have a factor of $G_N$. Finally, the terms in $\mathcal{L}_3^{(b)}$ give a three-point vertex involving one graviton leg, which will give corrections to the two-point function, again via loop corrections. Additionally, this term will give the leading behavior of the four-point function. Notice that the graviton exchange diagram will dominate over an exchange of $\check{\mathcal{X}}$ because the former will be proportional to $G_N/a$, while the latter will be proportional to $G_N e^{2\psi_0}$ and we are assuming that $a \ll e^{-2\psi_0}$. In addition, there are two type of diagrams that enter at the same order as the exchange of $\check{\mathcal{X}}$: i) a graviton exchange but with one of the legs sourced by a $\mathcal{Y}$ term, and ii) the four point contact diagram. For details, see the diagrams in Figs. 1 and 2.

## 3.5 Two-point functions at order $\mathcal{O}(G_N^0)$

Let us consider the first two type of diagrams in Fig. 1, which include the first effects of $\mathcal{Y}$ interacting with $\check{\mathcal{X}}$. Details of this calculation are presented in App. B; here we merely state

---

[7]We are being slightly imprecise with the treatment of the dimensionful parameter $a$. Here and in subsequent expressions we are always measuring $a$ relative to the AdS$_2$ boundary cutoff, i.e., we have $\mathcal{O}(ar_c e^{2\psi_0}) \ll 1$ as $r_c$ approaches the boundary. For brevity we omit the UV cutoff $r_c$, but it is implied in this discussion.

the results.

The first diagram in Fig. 1 is the free one, which leads to the standard CFT result,

$$\langle \mathcal{O}_{\check{\mathcal{X}}}(u_1)\mathcal{O}_{\check{\mathcal{X}}}(u_2)\rangle_\beta = D\left[\frac{\pi}{\beta\sin(\frac{\pi u_{12}}{\beta})}\right]^{2\Delta}, \tag{57}$$

with

$$D = \frac{(2\Delta-1)\Gamma[\Delta]}{\sqrt{\pi}\Gamma[\Delta-\frac{1}{2}]}, \qquad \Delta \equiv \Delta_{\mathcal{X}} = \frac{1}{2}\left(1 + 5\sqrt{\frac{1+\frac{28}{5}q}{1+12q}}\right). \tag{58}$$

The second diagram gives the leading order correction that arises from the coupling to the dilaton. The result of this diagram is given in (201) and, when combined with the free result, yields (202). We transcribe the final result here:

$$\langle \mathcal{O}_{\check{\mathcal{X}}}(u_1)\mathcal{O}_{\check{\mathcal{X}}}(u_2)\rangle_\beta = \left[\frac{\pi}{\beta\sin(\frac{\pi u_{12}}{\beta})}\right]^{2\Delta}\left[D + \frac{\tilde{D}a\beta^2}{2\pi^2}\left(2 + \pi\frac{1-2u_{12}/\beta}{\tan(\frac{\pi u_{12}}{\beta})}\right)\right]. \tag{59}$$

There is a new parameter, $\tilde{D}$, which we defined in (189) in terms of generic cubic couplings, and it was first reported for a generic cubic interaction in [7]. Specializing this formula for our IR theory in (55), we obtain:

$$\tilde{D} = -\frac{K_{\mathcal{Y}\check{\mathcal{X}}\check{\mathcal{X}}}e^{2\psi_0}}{2\ell_2^2(1+2q)^2(1+12q)}\left[(5+40q+124q^2+368q^3)+(1+6q)(9+38q-80q^2)\right], \tag{60}$$

with $K_{\mathcal{Y}\check{\mathcal{X}}\check{\mathcal{X}}}$ given in (185). It can be checked that $D > 0$, but $\tilde{D}$ can have either sign depending on the value of $q$ (or $\Delta$). $K_{\mathcal{Y}\check{\mathcal{X}}\check{\mathcal{X}}}$ has a definite negative sign, but the term inside the brackets can be positive or negative: $\tilde{D} > 0$ for $q \lesssim 2.85$ and $\tilde{D} < 0$ for $q \gtrsim 2.85$. Finally, we recall that $\Delta$ decreases monotonically as $q$ increases ($q \in [0,\infty]$) with the following limits:

$$3 \geq \Delta \geq \frac{1}{6}(3+\sqrt{105}) \approx 2.208. \tag{61}$$

So $\check{\mathcal{X}}$ is irrelevant, but never dominates the IR dynamics, which is controlled by $\mathcal{Y}$. In terms of $\Delta$, it can be checked that $\tilde{D}$ becomes negative in the range

$$2.235 \gtrsim \Delta \gtrsim 2.208. \tag{62}$$

Notice that the change in sign occurs only in the presence of a cosmological constant, and hence is a feature of $AdS_5$ black holes. We will comment on the possible physical interpretation of this result in Sec. 5. For the moment, we remark that the point where $\tilde{D} = 0$ corresponds to a regular black hole solution, with finite angular momentum and mass, hence this change of sign is somewhat unexpected.

## 3.6 Gravitational backreaction

We will now study the imprint of the graviton on the correlation functions. These corrections will come from the cubic terms in the Lagrangian denoted by $\mathcal{L}_3^{(b)}$ in (43). The two ingredients needed to deal with these terms are the propagators (both for the scalar and the graviton), which we already described, and the cubic vertex involving one graviton. We will now derive this vertex, and then proceed to estimate the corrections on the correlation functions of interest.

### 3.6.1 Cubic vertex involving one graviton

In order to be able to use the propagator (52) we need to express the cubic terms (56) in the same gauge (45). We will work in $(t, z)$ coordinates here (to directly compare our results with those in [24]), so we define

$$h_{tt} = -\frac{\ell_2^2}{z^2}(h + g), \qquad h_{zz} = \frac{\ell_2^2}{z^2}(h + g). \tag{63}$$

With these definitions (56) becomes

$$\mathcal{L}_3^{(b)} = -\frac{z^2}{2\ell_2^2} g \left[ (\partial_t \check{\mathcal{X}})^2 + (\partial_z \check{\mathcal{X}})^2 \right] - \frac{m_{\mathcal{X}}^2}{2} h \check{\mathcal{X}}^2, \tag{64}$$

so the relevant part of the action is

$$S_{2D} = \int dt\, dz \sqrt{-g}\, \mathcal{L}_3^{(b)} = \int dt\, dz \left\{ -\frac{1}{2} g \left[ (\partial_t \check{\mathcal{X}})^2 + (\partial_z \check{\mathcal{X}})^2 \right] - \frac{\ell_2^2 m_{\mathcal{X}}^2}{2z^2} h \check{\mathcal{X}}^2 \right\}. \tag{65}$$

The first part coincides with the last term of equation (3.8) in [24]. In that paper they considered a massless field, which is why the last term was absent. In that case, the vertex is given by

$$V_{g\check{\mathcal{X}}^2}^{(m=0)} = -i(\partial_t^1 \partial_t^2 + \partial_z^1 \partial_z^2), \tag{66}$$

where the subscripts 1 and 2 refer to the two external $\check{\mathcal{X}}$ legs. In our case we have $m_{\mathcal{X}} \neq 0$, so the last term in (64) contributes to the vertex. This term is a bit problematic because it couples to $h$ instead of $g$. To deal with this term, we replace

$$h \rightarrow g - \frac{1}{2} z (\partial_z g) \tag{67}$$

and integrating by parts the second term we obtain:

$$-\int dt\, dz \left( \frac{\ell_2^2 m_{\mathcal{X}}^2}{2z^2} h \check{\mathcal{X}}^2 \right) = -\int dt\, dz \left[ \frac{\ell_2^2 m_{\mathcal{X}}^2}{4z^2} g \check{\mathcal{X}}^2 + \frac{\ell_2^2 m_{\mathcal{X}}^2}{2z} g \check{\mathcal{X}}(\partial_z \check{\mathcal{X}}) \right] + \text{total derivative}. \tag{68}$$

This gives a correction to the vertex, such that now

$$V_{g\check{\mathcal{X}}^2} = -i \left[ \partial_t^1 \partial_t^2 + \partial_z^1 \partial_z^2 + \frac{\ell_2^2 m_{\mathcal{X}}^2}{4z^2} + \frac{\ell_2^2 m_{\mathcal{X}}^2}{4z} (\partial_z^1 + \partial_z^2) \right]. \tag{69}$$

Together with the graviton propagator (52), and the propagators for the scalar $\check{\mathcal{X}}$ (bulk-to-bulk and bulk-to-boundary) one can then, in principle, evaluate the desired diagrams, shown in Figs. 1 and 2. In practice, it is preferable to deal with the effects of the graviton by implementing a suitable diffeomorphism, which should be equivalent to the diagrammatic approach we described. Indeed, this seems easier to implement as many of the integrals needed to compute the diagrams must be done numerically, even for the massless case [24].

### 3.6.2 Effects on the two- and four-point functions at order $\mathcal{O}(G_N)$

The leading correction to the two-point function at order $\mathcal{O}(G_N)$ is given by the third diagram in Fig. 1. The leading diagram contributing to the four-point function also involves a graviton exchange, and is shown at the left of Fig. 2. As explained above, we can obtain these contributions directly from the diagrams or, alternatively, we can calculate them from an appropriate diffeomorphism. We chose the latter approach, because the calculations can be tracked down

analytically, as shown in e.g. [7, 34]. The calculations are standard, so we relegate the details to App. B.3.

For the two-point function, we find that the leading order correction is of the form (213):

$$\langle \mathcal{O}_\chi(u_1)\mathcal{O}_\chi(u_2)\rangle_\beta^{(\text{grav})} = D\left[\frac{\pi}{\beta\sin(\frac{\pi u_{12}}{\beta})}\right]^{2\Delta}\langle\mathcal{C}(u_1,u_2)\rangle, \tag{70}$$

where

$$\langle\mathcal{C}(u_1,u_2)\rangle = \frac{\Delta}{2\pi C}\left[\frac{\pi}{\beta\sin(\frac{\pi u_{12}}{\beta})}\right]^2\left[2+4\Delta+\frac{2\pi u_{12}}{\beta}(\frac{2\pi u_{12}}{\beta}-2\pi)(\Delta+1)\right. \tag{71}$$
$$\left.+\left(\frac{2\pi u_{12}}{\beta}(\frac{2\pi u_{12}}{\beta}-2\pi)\Delta-4\Delta-2\right)\cos(\frac{2\pi u_{12}}{\beta})+2(\pi-\frac{2\pi u_{12}}{\beta})(2\Delta+1)\sin(\frac{2\pi u_{12}}{\beta})\right]$$

is a combination of two-point correlators of the Schwarzian mode and $C$ is a constant of proportionality appearing in the effective action, which in our case is given by

$$C = \frac{\ell_2^2 a}{\kappa_2^2} = \frac{\ell_2^2 a}{8\pi G_N}. \tag{72}$$

For the four-point exchange diagram one obtains (215), which we copy here:

$$\langle\mathcal{O}_\chi(u_1)\mathcal{O}_\chi(u_2)\mathcal{O}_\chi(u_3)\mathcal{O}_\chi(u_4)\rangle_\beta^{(\text{grav})} = \frac{D^2}{2}\frac{\langle\mathcal{B}(u_1,u_2)\mathcal{B}(u_3,u_4)\rangle}{\left[2\sin(\frac{\pi u_{12}}{\beta})\right]^{2\Delta}\left[2\sin(\frac{\pi u_{34}}{\beta})\right]^{2\Delta}}, \tag{73}$$

again in terms of a particular combination of correlators of the Schwarzian mode. Upon analytic continuation, one finds that the OTOC relevant to chaos (216) behaves at late times as

$$\langle\mathcal{B}(0,t)\mathcal{B}(0,t)\rangle \sim \frac{\beta\Delta^2}{C}e^{\frac{2\pi}{\beta}t}, \qquad (t\gg\beta), \tag{74}$$

with a Lyapunov exponent that saturates the chaos bound [35],

$$\lambda_L = \frac{2\pi}{\beta}. \tag{75}$$

As expected via the diagram analysis, both (70) and (73) are of order $\mathcal{O}(1/C)\sim\mathcal{O}(G_N/a)$.

# 4 Interpolation: from UV to IR

In this section we will discuss the imprint of the IR analysis on the asymptotically 5D region. Using the rotating five-dimensional black hole as the background solution, we will quantify how the near-horizon fluctuations from Sec. 3 propagate to the far region of the black hole. Our main emphasis will be on the deformations that we identify as the JT sector in the IR region.

## 4.1 Black hole background and near-horizon region

To start, recall that the five-dimensional metric we are considering is of the form

$$g_{\mu\nu}^{(5)}\mathrm{d}x^\mu\mathrm{d}x^\nu = e^{\psi+\chi}g_{ab}^{(2)}\mathrm{d}x^a\mathrm{d}x^b + R^2 e^{-2\psi+\chi}d\Omega_2^2 + R^2 e^{-2\chi}(\sigma^3+A)^2. \tag{76}$$

The appropriate values of the fields that define the black hole background are given by

$$R^2 e^{-2\chi_{\text{BH}}} = \frac{\hat{r}^2}{4\Xi} + \frac{ma^2}{2\Xi^2\hat{r}^2} \, ,$$

$$R^2 e^{-2\psi_{\text{BH}}} = \frac{\hat{r}^2}{4\Xi} e^{-\chi_{\text{BH}}} \, ,$$

$$A_{\text{BH}} = \frac{a}{2R^2\Xi}\left(\frac{\hat{r}^2}{\ell_5^2} - \frac{2m}{\hat{r}^2}\right)e^{2\chi_{\text{BH}}}\,\mathrm{d}\hat{t} \, , \tag{77}$$

and

$$\left(e^{\psi+\chi} g^{(2)}_{ab}\mathrm{d}x^a\mathrm{d}x^b\right)_{\text{BH}} = \frac{\mathrm{d}\hat{r}^2}{\Delta(\hat{r})} - \frac{\Delta(\hat{r})}{\Xi}e^{-2\psi_{\text{BH}}+3\chi_{\text{BH}}}\mathrm{d}\hat{t}^2 \, , \tag{78}$$

where we introduced

$$\Xi = 1 - \frac{a^2}{\ell_5^2} \, , \qquad \Delta(\hat{r}) = \Xi + \frac{\hat{r}^2}{\ell_5^2} - \frac{2m}{\hat{r}^2} + \frac{2ma^2}{\hat{r}^4} \, . \tag{79}$$

Here $m$ and $a$ are constants; they are related to the mass and angular momentum of the black hole via [36]

$$M = \frac{3\pi^2\ell_5^2}{4\kappa_5^2} + \frac{2\pi^2}{\kappa_5^2}\frac{m(4-\Xi)}{\Xi^3} \, , \qquad J = \frac{8\pi^2}{\kappa_5^2}\frac{ma}{\Xi^3} \, . \tag{80}$$

Note that the mass includes the contribution from the Casimir energy. This black hole is a special case of the solutions constructed in [31]. We generally assume the black hole is asymptotically $\text{AdS}_5$, but the asymptotically flat Myers-Perry black holes are special cases of our analysis. It is also worth noting that the angular momentum $J$ is related to $Q$ in (8) as

$$Q = -\frac{\kappa_2^2}{R^2}J \, . \tag{81}$$

The interpolation between the IR ($\text{AdS}_2$) and UV ($\text{AdS}_5/\text{Minkowski}_5$) region is simplest to discuss for the extremal black hole. At extremality, the parameters of the black hole obey

$$\frac{2a_0^2}{\ell_5^2} = 1 + 2\frac{\hat{r}_0^2}{\ell_5^2} - \sqrt{1 + 2\frac{\hat{r}_0^2}{\ell_5^2}} \, ,$$

$$\frac{2m_0}{\hat{r}_0^2} = 1 + \frac{\hat{r}_0^2}{\ell_5^2} + \sqrt{1 + 2\frac{\hat{r}_0^2}{\ell_5^2}} \, , \tag{82}$$

where $\hat{r}_0$ satisfies $\Delta(\hat{r}_0) = 0$ and $\Delta(\hat{r}_0)' = 0$; the subscript "0" here denotes that these are the extremal values of the appropriate parameters. The coordinate transformation that encodes the decoupling limit is

$$\hat{r} = \hat{r}_0 + \lambda r \, ,$$

$$\hat{t} = \frac{\lambda_0}{\lambda}t \, ,$$

$$\hat{\psi} = \hat{\psi}_{\text{IR}} + \frac{\Omega_0}{\lambda}t \, , \tag{83}$$

with

$$\lambda_0^2 \equiv \Xi\ell_2^4 e^{4\psi_0-\chi_0} \, , \qquad \Omega_0 \equiv \frac{\ell_5}{R^2}\sqrt{2q(1+8q)}e^{2\chi_0}\lambda_0 \, . \tag{84}$$

The resulting near-horizon geometry obtained by using (83) on the AdS$_5$ black hole, and taking $\lambda \to 0$, is given by

$$g^{(5)}_{\mu\nu} dx^\mu dx^\nu = e^{\chi_0 + \psi_0} \bar{g}_{ab} dx^b dx^a + R^2 e^{-2\psi_0 + \chi_0} d\Omega_2^2 + e^{-2\chi_0} \left( \sigma^3_{\text{IR}} + \bar{A}_t dt \right)^2 + O(\lambda) , \qquad (85)$$

with $\sigma^3_{\text{IR}} = d\hat{\psi}_{\text{IR}} + \cos\theta d\phi$. This is precisely the IR solution in (16)-(17), with an AdS$_2$ metric as in (29). In relation to the notation of Sec. 3.1 we have

$$\frac{\hat{r}_0^2}{\ell_5^2} = 4q \frac{(1 + 6q)}{(1 + 4q)^2} . \qquad (86)$$

Further details about the near-horizon geometry of the black hole are presented in App. A, as well as the generalization to near-extremal black holes. In what follows, one of our main goals is to interpolate, via the decoupling limit (83), between linear perturbations on (76) and those on top of (85).

## 4.2 Recovering the JT sector

From the perspective of the near-horizon (IR) geometry, the elementary role of the JT field is to deviate the black hole away from extremality [4, 7]. That is, it increases the mass (or angular momentum) while keeping the angular momentum (or mass) fixed.[8] This change in parameters is done while preserving the topology and isometries of the event horizon, which in the context of our black hole solution means that we are preserving the spherical and rotational symmetry of the black hole. For this reason, it is expected that outside the near-horizon region the JT field should be identified with a perturbation that changes the conserved quantities of the black hole plus a diffeomorphism (to preserve a choice of gauge). This has been illustrated for four-dimensional black holes in [23, 37] and here we will carry out a similar analysis for the five-dimensional solution.

### 4.2.1 Linear perturbations around the black hole background

Around the black hole background we will consider linear perturbations of the form

$$g^{(5)} = g^{(5)}_{\text{BH}} + \epsilon \, \delta g^{(5)} , \qquad (87)$$

with

$$\delta g^{(5)} \equiv \delta_{M,J} g^{(5)}_{\text{BH}} + \mathcal{L}_\zeta g^{(5)}_{\text{BH}} , \qquad (88)$$

and $\delta_{M,J}$ denotes a change of mass or angular momentum (or both) of the black hole. The parameter $\epsilon$ in (87) is simply controlling the order, as in (38), which will be kept linear in this section. We also included a Lie derivative as part of the perturbations. In particular, we will consider in (88) single-valued diffeomorphisms that preserve the form of (76), which are

$$\zeta^\mu \partial_\mu = \zeta^a \partial_a + \zeta^{\hat{\psi}} \partial_{\hat{\psi}} . \qquad (89)$$

Here the components of $\zeta^\mu$ only depend on the two-dimensional coordinates $x^a$.[9] In the next few steps we will reorganize the information in (88)-(89) so its effect on the two-dimensional fields in (76) is more explicit.

---

[8]In general, interpreting the JT deformation as an increase of mass or another charge of the black hole depends on how the nearly-AdS$_2$ is embedded in a higher-dimensional space and possible ways of taking a decoupling limit. For our specific 5D solution the straightforward interpretation is to view the nearly-AdS$_2$ deformation as increasing the mass, since $J$ is related to $Q$ via (81), and we are holding $Q$ fixed in Sec. 3.

[9]Notice that in (89) one could include additional vector fields that would preserve (76). These are the Killing vectors of the $S^2$, but they won't affect any of the subsequent analysis here since they don't induce a change on the two-dimensional variables: the fields $\psi$, $\chi$, $A$ and $g_{ab}^{(2)}$.

It is convenient to introduce the notation

$$\boldsymbol{\delta} \equiv \delta - \delta_{M,J} \, , \tag{90}$$

to single out the portion of the transformation in (88) that depends on $\zeta$. Focusing first on the angular components of $g^{(5)}_{\mu\nu}$, the left and right hand side of (88) lead to the following relations

$$\zeta^{\hat{r}} \partial_{\hat{r}} e^{-2\chi_{\text{BH}}} = \boldsymbol{\delta} e^{-2\chi} \, , \tag{91}$$

$$\zeta^{\hat{r}} \partial_{\hat{r}} e^{-2\psi_{\text{BH}}+\chi_{\text{BH}}} = \boldsymbol{\delta} e^{-2\psi+\chi} \, , \tag{92}$$

where we used the fact that the black hole is a stationary solution. These equations relate $\zeta^{\hat{r}}$ to the fluctuations of $\psi$ and $\chi$; more importantly, for perturbations of the form (88) we find the relation

$$\partial_{\hat{r}} \chi_{\text{BH}} \boldsymbol{\delta}\psi = \partial_{\hat{r}} \psi_{\text{BH}} \boldsymbol{\delta}\chi \, , \tag{93}$$

which is one of the characteristic features of perturbations induced by a Lie derivative. The only other two independent relations one gets from (88) are

$$\zeta^c \partial_c g^{(5)}_{\hat{\psi}a} + g^{(5)}_{\hat{\psi}\hat{\psi}} \partial_a \zeta^{\hat{\psi}} + g^{(5)}_{\hat{\psi}c} \partial_a \zeta^c = \boldsymbol{\delta} g^{(5)}_{\hat{\psi}a} \, ,$$
$$\zeta^c \partial_c g^{(5)}_{ab} + g^{(5)}_{ac} \partial_b \zeta^c + g^{(5)}_{bc} \partial_a \zeta^c + g^{(5)}_{\hat{\psi}a} \partial_b \zeta^{\hat{\psi}} + g^{(5)}_{\hat{\psi}b} \partial_a \zeta^{\hat{\psi}} = \boldsymbol{\delta} g^{(5)}_{ab} \, . \tag{94}$$

Here and in the following, we omitted the subscript "BH" for compactness, but it is implied that the metric components are those of the black hole. Using (91) and (93) we can simplify these expressions, reducing them to the following expression. First, from the components along $x^a$ in (94) we find

$$\zeta^c \partial_c g^{(2)}_{ab} + g^{(2)}_{ac} \partial_b \zeta^c + g^{(2)}_{bc} \partial_a \zeta^c = \boldsymbol{\delta} g^{(2)}_{ab} \, , \tag{95}$$

which reflects which portion of the five-dimensional diffeomorphism acts as a Lie derivative on the two-dimensional metric. And, not surprisingly, we also obtain

$$\zeta^c \partial_c A_a + A_c \partial_a \zeta^c + \partial_a \zeta^{\hat{\psi}} = \boldsymbol{\delta} A_a \, , \tag{96}$$

describing a Lie derivative plus a $U(1)$ gauge transformation, with $\zeta^{\hat{\psi}}$ the gauge function, acting on the vector field $A_a$.

Finally, we should make use of some residual gauge freedom. Without loss of generality, we will set

$$\delta g^{(2)}_{\hat{t}\hat{r}} = 0 \, . \tag{97}$$

Using the properties of the black hole background, the components of (95) then lead to

$$g^{(2)}_{\hat{r}\hat{r}} \partial_{\hat{t}} \zeta^{\hat{r}} + g^{(2)}_{\hat{t}\hat{t}} \partial_{\hat{r}} \zeta^{\hat{t}} = 0 \, , \tag{98}$$

$$2\sqrt{g^{(2)}_{\hat{r}\hat{r}}} \partial_{\hat{r}} \left( \sqrt{g^{(2)}_{\hat{r}\hat{r}}} \zeta^{\hat{r}} \right) = \boldsymbol{\delta} g^{(2)}_{\hat{r}\hat{r}} \, , \tag{99}$$

$$\zeta^{\hat{r}} \partial_{\hat{r}} g^{(2)}_{\hat{t}\hat{t}} + 2g^{(2)}_{\hat{t}\hat{t}} \partial_{\hat{t}} \zeta^{\hat{t}} = \boldsymbol{\delta} g^{(2)}_{\hat{t}\hat{t}} \, . \tag{100}$$

The first equation will be used to eliminate $\zeta^{\hat{t}}$ from subsequent equations; the second equation determines the radial profile of $\zeta^{\hat{r}}$. The last equation, after using (98), can be written as

$$\partial_{\hat{r}} \left( \frac{\partial_{\hat{r}} g^{(2)}_{\hat{t}\hat{t}}}{g^{(2)}_{\hat{t}\hat{t}}} \zeta^{\hat{r}} \right) - 2\frac{g^{(2)}_{\hat{r}\hat{r}}}{g^{(2)}_{\hat{t}\hat{t}}} \partial_{\hat{t}}^2 \zeta^{\hat{r}} = \partial_{\hat{r}} \left( \frac{\boldsymbol{\delta} g^{(2)}_{\hat{t}\hat{t}}}{g^{(2)}_{\hat{t}\hat{t}}} \right) \, . \tag{101}$$

Recall that $\zeta^{\hat{r}}$ is related to $\boldsymbol{\delta}\chi$ and $\boldsymbol{\delta}\psi$ via (91)-(93), which will allow us to interpret (99) and (101) as equations for $\boldsymbol{\delta}\chi$ and $\boldsymbol{\delta}\psi$.

### 4.2.2 Decoupling limit

Next, we will determine the profiles of $\zeta^a$ that have a well-defined decoupling limit that can be matched to the IR perturbations in Sec. 3.1. In particular, we want to relate the equations that determine the UV perturbations

$$\delta\chi\,, \quad \delta\psi\,, \quad \delta g_{ab}^{(2)}\,, \tag{102}$$

to the equations that govern the IR perturbations in (20),

$$\mathcal{X}\,, \quad \mathcal{Y}\,, \quad h_{ab}\,, \tag{103}$$

via the decoupling limit (83). This means that the UV perturbations should be thought of as an expansion in $\lambda$, which we assume to be well-defined as we reach the near-horizon region of the black hole. For the scalar perturbations, this implies that they admit a Taylor expansion of the form

$$\delta\chi = (\delta\chi)_0 + \sum_{n=1}^{\infty} (\delta\chi)_n \lambda^n\,, \qquad \delta\psi = (\delta\psi)_0 + \sum_{n=1}^{\infty} (\delta\psi)_n \lambda^n\,, \tag{104}$$

which leads to perturbations that are finite even if we take the strict limit $\lambda \to 0$. The two-dimensional metric fluctuations scale as

$$\delta g_{\hat{r}\hat{r}}^{(2)} = O(\lambda^{-2})\,, \qquad \delta g_{\hat{t}\hat{t}}^{(2)} = O(\lambda^2)\,. \tag{105}$$

This assures that the line element, including the linear metric perturbation, is finite as we approach the IR background via (83).

Note that it is possible to also identify the perturbation parameter $\epsilon$ in (38) and (87) with the decoupling parameter $\lambda$, which is discussed partially in [18] for these black holes. The technical aspects of this discussion follow in a straightforward way if we set $\epsilon \sim \lambda$, but as presented here the discussion is more general.

The simplest relation between the IR and UV equations can be established from (93). Using the background values of the black hole in (77), and taking the decoupling limit of it, we obtain

$$\boldsymbol{\delta}\chi = -\frac{4q}{1+2q}\boldsymbol{\delta}\psi\,. \tag{106}$$

which agrees with (36) provided we identify $\boldsymbol{\delta}\chi$ with $\mathcal{X}$, and $-2e^{-2\psi_0}\boldsymbol{\delta}\psi$ with $\mathcal{Y}$. However, we should be careful that, as defined in (90), the symbol $\boldsymbol{\delta}$ encodes the linear perturbation and a mass change; and we have not specified the interplay between a change in mass (or angular momentum) with the decoupling limit. Without loss of generality, we will focus only on changing the mass of the black hole, while keeping the angular momentum fixed. In this case we have

$$\boldsymbol{\delta}\chi = \delta\chi - \left(\frac{\partial\chi_{\text{BH}}}{\partial M}\right)_J \delta M\,. \tag{107}$$

As we take an extremal, or near-extremal limit, the term

$$\left(\frac{\partial\chi_{\text{BH}}}{\partial M}\right)_J = \frac{\kappa_5^2}{16\pi^2}\frac{(1+14q+16q^2-256q^3)}{(1+2q)(1+6q)(3+16q)}e^{2\chi_0} + O(\lambda)\,, \tag{108}$$

remains finite, and a similar expression holds for $\boldsymbol{\delta}\psi$. As we discuss the remaining equations, we will impose conditions on $\delta M$, assuming that $\delta M \sim (\delta M)_n \lambda^n$ with $n > 1$. Then we see that in the strict $\lambda \to 0$ limit, $\boldsymbol{\delta}\chi = (\delta\chi)_0 + O(\lambda)$ and $\boldsymbol{\delta}\psi = (\delta\psi)_0 + O(\lambda)$, and hence (107) reduces to

$$(\delta\chi)_0 = -\frac{4q}{1+2q}(\delta\psi)_0\,, \tag{109}$$

in complete agreement with (36). Note that we will also have a simple relation relation among $\zeta^{\hat{r}}$ and the fluctuations: from (91)-(92) we have

$$(\zeta^{\hat{r}})_0 = -2\ell_5 \frac{\sqrt{q(1+6q)}}{1+2q}(\delta\psi)_0 \,, \tag{110}$$

where we are expanding

$$\zeta^{\hat{r}} = \sum_{n\geq 0}(\zeta^{\hat{r}})_n \lambda^n \,.$$

Hence, in the IR, the variables $(\zeta^{\hat{r}})_0$, $(\delta\chi)_0$, and $(\delta\psi)_0$ are related in a simple manner by constant background factors.

Next we need to analyse (99) and (101). For these equations the background metric components $g_{ab}^{(2)}$ have a non-trivial scaling with $\lambda$ which we now quantify. From the definitions in (78)-(79) we have

$$g_{\hat{r}\hat{r}}^{(2)} = \bar{g}_{rr}\lambda^{-2} + O(\lambda^{-1}) \,, \qquad g_{\hat{t}\hat{t}}^{(2)} = \bar{g}_{tt}\frac{\lambda^2}{\lambda_0^2} + O(\lambda) \,, \tag{111}$$

where $\bar{g}_{ab}$ is the two-dimensional AdS metric in (29). The effect of changing the mass, while keeping the angular momentum fixed, reads

$$\begin{aligned}
\delta_M g_{\hat{r}\hat{r}}^{(2)} &= \lambda^{-4}\frac{\ell_2^4}{r^4}m\,\delta M + O(\lambda^{-3}) \,, \\
\delta_M g_{\hat{t}\hat{t}}^{(2)} &= \frac{m}{\Xi}e^{-4\psi_0+\chi_0}\delta M + O(\lambda) \,,
\end{aligned} \tag{112}$$

with

$$m \equiv \frac{\kappa_5^2}{4\pi^2 R^2}\frac{1+6q}{1+2q}e^{\psi_0+3\chi_0} \,. \tag{113}$$

Replacing (111) and (112) in (99) and (101), we see that the leading $\lambda$ behavior of these equations becomes

$$2\lambda^{-3}\sqrt{\bar{g}_{rr}}\partial_r\left(\sqrt{\bar{g}_{rr}}(\zeta^{\hat{r}})_0\right) = \delta g_{\hat{r}\hat{r}}^{(2)} - \lambda^{-4}\frac{\ell_2^4}{r^4}m\,\delta M \,, \tag{114}$$

and

$$\lambda^{-2}\partial_r\left(\frac{\partial_r\bar{g}_{tt}}{\bar{g}_{tt}}(\zeta^{\hat{r}})_0\right) - 2\lambda^{-2}\frac{\bar{g}_{rr}}{\bar{g}_{tt}}\partial_t^2(\zeta^{\hat{r}})_0 = \lambda^{-3}\lambda_0^2\partial_r\left(\frac{\delta g_{\hat{t}\hat{t}}^{(2)}}{\bar{g}_{tt}}\right) - \lambda^{-3}\ell_2^4\partial_r\left(\frac{1}{\bar{g}_{tt}}\right)m\,\delta M \,. \tag{115}$$

What we observe from these equations is the following. First, given the scaling with $\lambda$ in (105), $\delta g_{\hat{r}\hat{r}}^{(2)}$ and $\delta g_{\hat{t}\hat{t}}^{(2)}$ drop at leading order in these two equations. Second, if we take $\delta M \sim \lambda$ then the mass change survives in both (114) and (115); the resulting equations we obtain are

$$\begin{aligned}
\partial_r\left(\frac{1}{r}\partial_t(\zeta^{\hat{r}})_0\right) &= 0 \,, \\
r^2\partial_r^2(\zeta^{\hat{r}})_0 + r\partial_r(\zeta^{\hat{r}})_0 - (\zeta^{\hat{r}})_0 &= 0 \,,
\end{aligned} \tag{116}$$

where we took a time and radial derivative to eliminate $\delta M$ in (114), and we are also using (29) for $\bar{g}_{ab}$. Using these results, we find from (115)

$$\partial_t^2(\zeta^{\hat{r}})_0 - r^3\partial_r(\zeta^{\hat{r}})_0 + r^2(\zeta^{\hat{r}})_0 = 0 \,. \tag{117}$$

Equations (116)-(117) are the components of the JT equation (37) for $AdS_2$ in Poincaré coordinates. Since $(\zeta^{\hat{r}})_0$ has a simple relation to $(\delta\chi)_0$ and $(\delta\psi)_0$ via (109)-(110), we have captured the dynamics of the JT sector, matching perfectly the IR relations in Sec. 3.1. In particular we have

$$\mathcal{X} = (\delta\chi)_0 \,, \qquad \mathcal{Y} = -2e^{-2\psi_0}(\delta\psi)_0 \,, \tag{118}$$

which are tied to the radial component of the UV diffeomorphism via (110). It is important to stress that although the JT sector is recovered from a change of mass plus a Lie derivative in the UV, the decoupling limit tampers with this origin: in the IR, the JT sector is not accounted for by diffeomorphisms plus a change of mass on the $AdS_2$ background.

One portion of the IR equations that we have not addressed so far is the equation for the two-dimensional metric perturbations: how to obtain (31) from the UV equations (99)-(101). The metric perturbations are subleading in (115) and (114), and to extract the appropriate IR equations one simply needs to quantify subleading terms in $\lambda$ appearing in these equations. A tedious, but straightforward analysis, shows agreement with (31).

Finally, one should also inspect (96) to ensure that the gauge field $A_a$ has a well-defined behaviour as we take the decoupling limit. Since $\zeta^{\hat{r}}$ and $\zeta^{\hat{t}}$ are already determined from the JT sector, the requirements one would deduce from (96) will fix the decoupling behaviour of $\zeta^{\hat{\psi}}$. The explicit form of $\zeta^{\hat{\psi}}$ will not be needed for our subsequent analysis here, since our focus is only on the JT sector. But it would be interesting to decode its possible behaviour in future work: as illustrated recently in [29], one can impose different boundary conditions on gauge fields in the IR, and this leads to different holographic interpretations.

## 4.3  UV imprint of $\delta_M g^{(5)}_{\text{BH}} = 0$

In the prior analysis we discussed how the UV equations governing the perturbations of the form (88) reduce to the equations of motion that govern the JT sector in accordance to our results in Sec. 3.1. Now, we want to explicitly construct solutions to the UV equations and quantify how they behave from the perspective of the $AdS_5$ boundary.

To simplify the analysis we will solve the system when $\delta_M g^{(5)}_{\text{BH}} = 0$, illustrating the role of the diffeomorphisms in (87)-(88). We will also further impose the radial gauge $\delta g^{(5)}_{\hat{r}\hat{r}} = 0$, which will ease a comparison with the Fefferman-Graham expansion of $AdS_5$ – although other choices are of course perfectly reasonable. In this setup, equations (99) and (101) simplify to

$$\partial_{\hat{r}}\left(\sqrt{e^{\psi_{\text{BH}}+\chi_{\text{BH}}} g^{(2)}_{\hat{r}\hat{r}}} \, \zeta^{\hat{r}}\right) = 0 \,,$$
$$\partial_{\hat{r}}\left(\frac{\partial_{\hat{r}} g^{(2)}_{\hat{t}\hat{t}}}{g^{(2)}_{\hat{t}\hat{t}}}\zeta^{\hat{r}}\right) - 2\frac{g^{(2)}_{\hat{r}\hat{r}}}{g^{(2)}_{\hat{t}\hat{t}}}\partial_{\hat{t}}^2\zeta^{\hat{r}} = \partial_{\hat{r}}\left(\frac{\delta g^{(2)}_{\hat{t}\hat{t}}}{g^{(2)}_{\hat{t}\hat{t}}}\right) \,. \tag{119}$$

The first equation is straightforward to solve: using (78), we obtain

$$\zeta^{\hat{r}} = d_1(\hat{t})\sqrt{\Delta(\hat{r})} \,. \tag{120}$$

Here $d_1(\hat{t})$ is not completely arbitrary: we want this function to have a well-defined decoupling limit that matches $(\zeta^{\hat{r}})_0$ as $\lambda \to 0$, and hence complies with (116)-(117). This constrains the behavior of this function to be of the form

$$d_1(\hat{t}) = \frac{\lambda_0}{\lambda}(d_- + d_0 t) + O(\lambda^0) \,, \tag{121}$$

where we are using the relations in (83); here $d_-$ and $d_0$ are constants. Notice that we are expressing $d_1(\hat{t})$ in terms of the IR time $t$; in this form, its scaling with $\lambda$ is most transparent.

This profile for $\zeta^{\hat{r}}$ then, via (110) and (118), leads to the solution of the JT equations in (34) with

$$\mathcal{Y}(r,t) = c^- r + c^0 r t \,, \tag{122}$$

which implies that we are only turning on two of the three $sl(2)$ charges in the IR. From the second equation in (119), we find

$$\delta g_{\hat{t}\hat{t}}^{(2)} = \partial_{\hat{r}} g_{\hat{t}\hat{t}}^{(2)} \zeta^{\hat{r}} + g_{\hat{t}\hat{t}}^{(2)} d_2(\hat{t}) - 2 g_{\hat{t}\hat{t}}^{(2)} \int \mathrm{d}\hat{r} \left( \frac{g_{\hat{r}\hat{r}}^{(2)}}{g_{\hat{t}\hat{t}}^{(2)}} \partial_{\hat{t}}^2 \zeta^{\hat{r}} \right) \,. \tag{123}$$

The function $d_2(\hat{t})$ is also not completely arbitrary: by demanding that (123) scales as (105) in the IR, we find

$$d_2(\hat{t}) = -\frac{2}{\ell_2} \frac{\lambda_0}{\lambda} (d_- + d_0 t) e^{-\frac{1}{2}\psi_0 - \frac{1}{2}\chi_0} + O(\lambda^0) \,. \tag{124}$$

Next, let us quantify the imprint of (120) in the far $\mathrm{AdS}_5$ region. Here it will be important to make our comparisons relative to the background metric for the black hole (76)-(78), which as $\hat{r} \to \infty$ behaves like

$$g_{\mu\nu}^{(5)} \mathrm{d}x^\mu \mathrm{d}x^\nu = \ell_5^2 \frac{\mathrm{d}\hat{r}^2}{\hat{r}^2} + \frac{\hat{r}^2}{\Xi} \left[ -\frac{\mathrm{d}\hat{t}^2}{\ell_5^2} + \frac{1}{4} d\Omega_2^2 + \frac{1}{4} \left( \sigma_3 + 2\frac{a}{\ell_5^2} \mathrm{d}\hat{t} \right)^2 \right] + \cdots \,, \tag{125}$$

where the dots are subleading terms in $\hat{r}$. The term in square brackets is the boundary metric, which in this case is a co-rotating frame relative to a static frame defined by global $\mathrm{AdS}_5$.

Writing out the effect of (120) on the metric components of (87), the effect on the size of $S^2$ and the fibration are

$$\delta g_{\hat{\psi}\hat{\psi}}^{(5)} = R^2 \zeta^{\hat{r}} \partial_{\hat{r}} e^{-2\chi_{\text{BH}}} = \frac{\hat{r}^2}{2\ell_5 \Xi} d_1(\hat{t}) + \cdots \,,$$

$$\delta g_{\hat{\theta}\hat{\theta}}^{(5)} = R^2 \zeta^{\hat{r}} \partial_{\hat{r}} e^{-2\psi_{\text{BH}} + \chi_{\text{BH}}} = \frac{\hat{r}^2}{2\ell_5 \Xi} d_1(\hat{t}) + \cdots \,, \tag{126}$$

where we are expanding these expressions for large $\hat{r}$, i.e. near the $\mathrm{AdS}_5$ boundary. From (120) and (126) the effect of the radial diffeomorphism is clear: we are doing a Weyl transformation of the $\mathrm{AdS}_5$ boundary metric, with the Weyl factor being $(1 + \epsilon \frac{2d_1(t)}{\ell_5})$. This is also reflected on the time components of the boundary metric, although some care is needed since the boundary is squashed. For the time components, there is also a re-scaling by $(1 + \epsilon \frac{2d_1(t)}{\ell_5})$, and in addition $d_2(\hat{t})$ acts as an extra reparametrization of the boundary time. To be more explicit, we find

$$\delta g_{\hat{t}\hat{t}}^{(5)} = -\frac{\hat{r}^2}{\Xi \ell_5^2} \left( \frac{2d_1(\hat{t})}{\ell_5} + d_2(\hat{t}) \right) + \cdots \,, \tag{127}$$

and an analogous expression for $\delta g_{\hat{t}\hat{\psi}}^{(5)}$. Note that in this expression we have not included the contribution from $\zeta^{\hat{\psi}}$, which would act as a shift of $\sigma_3$. It is rather unexpected and interesting that the diffeomorphism that behaves smoothly in the IR corresponds to conformal transformations of the $\mathrm{AdS}_5$ boundary metric. However, if we take $\lambda \to 0$ in (126) and (127) while keeping $(\hat{r}, \hat{t}, \hat{\psi})$ fixed, the constant pieces of $d_1(\hat{t})$ and $d_2(\hat{t})$ blow up, so these expressions are sensitive to the decoupling parameter $\lambda$ and one should be careful with the limits. The most simple outcome is to just set $d_- = 0$, and hence this part of the JT sector does not extend to the exterior geometry.

It is interesting to compare this to e.g. [30] where the interpolation is between $\mathrm{AdS}_3$ and $\mathrm{AdS}_5$. In that case, the boundary gravitons of $\mathrm{AdS}_5$ simply contribute to the UV boundary

stress tensor, which is a subleading effect in comparison to (126). Also, there the interpolation is in reverse: the map in [30] is between large diffeomorphisms of $AdS_3$ to propagating gravitational modes in $AdS_5$, while here the diffeomorphisms lay on the $AdS_5$ region. It would be interesting to confirm that our conclusion here is not an artifact of a choice of gauge, but a robust interpretation of the JT sector in the UV of $AdS_5$.

The imprint is different if the UV region is instead asymptotically flat. For the case of zero cosmological constant, $\ell_5 \to \infty$, we can follow a similar procedure to study the behaviour of $\zeta$ in the asymptotic region. In this case the large $\hat{r}$ behaviour is

$$g^{(5)}_{\mu\nu} dx^\mu dx^\nu = -d\hat{t}^2 + d\hat{r}^2 + \frac{\hat{r}^2}{4} d\Omega_2^2 + \frac{\hat{r}^2}{4} \sigma_3^2 + \cdots , \qquad (128)$$

which is just $\mathbb{R}^{1,4}$. The procedure is completely analogous; one can simply use

$$\zeta^{\hat{r}}_{\Lambda_5=0} = d_1(\hat{t}) \sqrt{\lim_{\ell_5 \to \infty} \Delta(\hat{r})} . \qquad (129)$$

As a representative example of the behavior we find, consider

$$\delta\left(g^{(5)}_{\hat{\psi}\hat{\psi}}\right)_{\Lambda_5=0} = R^2 \zeta^{\hat{r}}_{\Lambda_5=0} \, \partial_{\hat{r}} e^{-2\chi_{\text{BH}}} = \frac{\hat{r}}{2} d_1(\hat{t}) + \cdots , \qquad (130)$$

which is subleading in the large $\hat{r}$ expansion. The other components of the Lie derivative of the metric are as well subleading in a large $\hat{r}$ expansion relative to (128). It would be interesting to investigate from an asymptotic symmetry group analysis of $\mathbb{R}^{1,4}$ if the falloffs here give rise to well-defined Iyer-Wald charges. Although they are subleading, it is possible that these falloffs are still singular transformations which forces one to set $d_{1,2}(\hat{t})$ equal to zero. This scenario would imply that only the mass deformation piece of the JT sector is physical from the perspective of the UV, and the other two $sl(2)$ modes in $\mathcal{Y}$ would be unphysical. This is not necessarily a negative outcome, and we refer to [7] for a discussion on why part of the $sl(2)$ modes should be gauged from the perspective of the Schwarzian action.

## 5 Discussion

In this last section we will discuss the implications of our findings, with particular emphasis on future directions that are interesting to explore. In a nutshell, our main results are two-fold. In Sec. 3, we focused on interacting aspects of the nearly-$AdS_2$ region; we found a region of the parameter $q$ for which $\tilde{D}$, which parametrizes the correction to the two-point function, flips sign. This occurs only when the black hole is embedded in $AdS_5$. In Sec. 4 we have explicitly shown how a perturbation in the IR propagates all the way in the UV region; in other words, we found the effect of the solutions for $\mathcal{X}, \mathcal{Y}, h_{ab}$ on the complete black hole solution. In both of these directions we have found precise quantitative differences when the black hole is embedded in $AdS_5$ versus Minkowski$_5$, which are worth exploring further.

Before delving into the detailed discussion of our results, let us mention that our formalism can be applied in several other contexts. For example, considering charged black holes such as those in [38] should be feasible, and an interesting case to study the interplay of charge and rotation. Another direction is to study rotating four-dimensional black holes, including Kerr-Newman and Kaluza-Klein black holes [39], although the near-horizon perturbations in that case assume a much more complicated form [23, 40].[10] Moreover, the methods in this

---

[10]To overcome the challenges of more general rotating solutions, one could consider treating rotation as a perturbative parameter as was done in [17, 20, 22].

paper may possibly be used in the context of dS$_2$ JT gravity, to understand parts of the space of near-dS$_2$ deformations (perhaps carefully considering appropriate analytic continuations). One could consider the near-Nariai limit of dS$_4$ black holes with small rotation parameter and investigate the effect of a small rotation parameter on the near-dS$_2$; see, for instance, [41] for a study of the interplay between the IR JT gravity and the UV 4D Schwarzschild-de Sitter. Work in this direction is in progress by some of us [42].

## 5.1 Stability properties of rotating AdS$_5$ black holes

A natural question to ask is what the imprint of the change of sign of $\tilde{D}$ in (60) is on the full UV solution. It could for instance be related to the thermodynamic stability and/or the quasinormal modes of the gravitational solutions. Many aspects of the thermodynamics and stability of rotating AdS$_5$ solutions have been examined in the past, and there are a few possible options that we analyzed.

First of all, for sufficiently low angular momentum, in the ensemble of fixed temperature and angular momentum $J$, five-dimensional black holes undergo a liquid-gas-like phase transition [43] similar to those found in [44,45]. This process manifests itself in a discontinuous first derivative of the free energy: at this point, small (with respect to the AdS radius) black holes turn into large ones. The phase transition happens at sufficiently low angular momentum, and the onset in our parametrization is for $q \sim 0.02$, which corresponds to a much smaller value of angular momentum with respect to the location where $\tilde{D}$ changes sign. Therefore we rule out a possible relation between the two phenomena. Similarly, a possible relation to an instability connecting rotating AdS$_5$ black holes and an AdS black ring is also likely to be excluded.[11] While in [47] AdS$_5$ thin black rings were constructed via approximate methods, recent studies seem to exclude extremal AdS$_5$ black rings in Einstein-Maxwell-$\Lambda$ theories [48,49]. The effective IR theory describing these objects would be of the same form as our effective AdS$_2$ description, due to a five-dimensional decomposition similar to that performed in Sec. 2. While other types of solutions (i.e. extremal black saturns) are not to be excluded, we are not in the position to claim that they will be directly relevant for our UV interpretation of the change in sign in $\tilde{D}$.

In [50,51] a master equation for the metric perturbations relevant for the study of stability of rotating five-dimensional black holes was provided. The subsequent analysis of AdS$_5 \times S^5$ black holes of [52] found two kinds of classical instabilities: the superradiant and the Gregory-Laflamme instabilities. The first kind of instability is caused by the wave amplification via superradiance, and by wave reflection due to the gravitational AdS potential. In App. A, we show that our (near-)extremal black holes are unstable against superradiance for any value of the extremal mass, and therefore this instability does not reflect the change of sign in $\tilde{D}$. The Gregory-Laflamme instability appears instead in the Kaluza-Klein modes of the internal manifold $S^5$. This instability affects a region of parameter space that is disconnected from the extremal black hole; therefore it is triggered by a (possibly) small departure from zero-temperature. Our analysis however is of a slightly different nature, because it does not involve any analysis of the KK modes of the internal manifold: the reduction to two dimensions is performed directly on the five-dimensional theory. It would be interesting nevertheless to investigate the possible relation between $\tilde{D}$ and this type of instability.

Finally, one could wonder if the change in sign of the correction to the two-point function could be reflected in the quasinormal modes (QNMs) spectrum of the black hole. The latter were investigated for instance in [53], where the shear viscosity and various other transport coefficients were computed holographically from Kerr-AdS$_5$ black holes, by computing the

---

[11]In [46] an instability of asymptotically flat black holes with large angular momentum ("ultraspinning" black holes) was detected. However, in our case the angular momentum parameter $a$ is bounded from above by $\ell_5$; therefore their analysis is not directly applicable.

QNMs associated with three sectors of decoupled perturbations (tensor, vector, scalar). It is known that for rotating black holes part of the frequency spectrum bifurcates near extremality into what is called "zero-damping modes" and "damped modes" (see for example [54]). It would be very interesting to try to formulate a more direct relation between these QNMs and the two-point functions of the 2D effective theory.

## 5.2 Holographic dual interpretation

It is also interesting to ask what a microscopic dual description of our gravitational system may look like. Several aspects of JT gravity are well described by a particular integrable limit of a class of SYK-like models [8–11], and for this reason it is worth placing our results in that context.

The original SYK-model is a zero dimensional quantum mechanics model of $N \gg 1$ Majorana fermions $\psi_i$ with all-to-all four fermion interactions:

$$H = -\sum_{j<k<l<m} J_{jklm}\psi_j\psi_k\psi_l\psi_m\,, \qquad \{\psi_j,\psi_k\} = \delta_{jk}\,. \tag{131}$$

The couplings $J_{jklm}$ here are independent random variables with zero mean, $\overline{J_{jklm}} = 0$, and fixed variance

$$\overline{J_{jklm}^2} = \frac{3!}{N^3}J^2\,, \tag{132}$$

where $J$ is a characteristic energy scale. A slight generalization that involves $q$ fermion interactions (for even $q$) is given by

$$H = (i)^{q/2}\sum_{j_1<\cdots<j_q} J_{j_1\ldots j_q}\psi_{j_1}\cdots\psi_{j_q}\,, \qquad \overline{J_{j_1\ldots j_q}^2} = \frac{(q-1)!}{N^{q-1}}J^2\,. \tag{133}$$

These are often referred to as the $q$SYK models and can be solved in the $N \to \infty$ limit using dynamical mean field theory. At infinite $q$, the system exhibits an emergent conformal symmetry and one recovers the standard two-point function (for bilinear, primary, $O(N)$ singlet operators) that normally follows from an AdS$_2$ fixed point,

$$G_c(u) = \frac{1}{2}\frac{1}{|\mathcal{J}u|^{2\Delta}}\,, \qquad \mathcal{J} \equiv \sqrt{q}\frac{J}{2^{\frac{q-1}{2}}}\,. \tag{134}$$

Interestingly, at finite $q$ one gets a series of corrections of the form [10],

$$G(u) = G_c(u)\left(1 - \frac{2}{q}\frac{1}{\mathcal{J}|u|} + \cdots\right)\,, \tag{135}$$

which, at finite temperature, acquire the same functional dependence as our result (59):

$$G(u) = G_c(u)\left[1 - \frac{2}{q}\frac{1}{\beta\mathcal{J}}\left(2 + \frac{\pi - 2\pi|u|/\beta}{\tan(\frac{\pi|u|}{\beta})}\right) + \cdots\right]\,. \tag{136}$$

We note, however, that the sign of the correction for this type of models is completely fixed, while in our case $\tilde{D}$ in (60) can be either positive or negative, depending on the value of $q$ that defines the extremal solution.

To understand this feature and possibly come up with a microscopic model dual to our two-dimensional system, we should look at the precise mapping between SYK and its bulk

dual [13]. It is known that the holographic dual to pure SYK should contain an infinite tower of massive particles dual to singlet operators of the form [12]

$$\mathcal{O}_n = \sum_{i=1}^{N} \psi_i \partial_u^{1+2n} \psi_i . \tag{137}$$

The standard AdS/CFT dictionary then dictates that each of these operators corresponds to a bulk scalar field $\phi_n$, with mass determined by its (IR) conformal dimension $\Delta_n$, according to $m_n^2 \ell_2^2 = \Delta_n(\Delta_n - 1)$. The couplings between these scalar fields could then be determined from various correlation functions of the composite operators $\mathcal{O}_n$ [13]. In particular, two-point functions $\langle \mathcal{O}_n(u_1) \mathcal{O}_n(u_2) \rangle$, or equivalently, fermion four-point functions, such as

$$\left\langle N^{-2} \sum_{i,j} \psi_i(u_1) \psi_i(u_1) \psi_j(u_2) \psi_j(u_2) \right\rangle , \tag{138}$$

suffice to determine the masses $m_i$. Likewise, three-point functions of $\mathcal{O}_n$, or fermion six-point functions, are enough to determine the bulk cubic couplings, e.g. $\frac{1}{\sqrt{N}} \lambda_{nmk} \phi_n \phi_m \phi_k$. Thus, it is clear that to come up with a microscopic model that can reproduce our corrected two-point function (59) we should modify the pure SYK to allow for more general fermion six-point functions. It should also allow for operators that reproduce the conformal dimension of our squashing mode $\mathcal{X}$, which is an irrational number (see (41)).

Models that could potentially lead to such modifications include those with the possible addition of extra fermion flavors [55, 56], or scalars, such as the supersymmetric generalizations of SYK and their extensions [57–59]. However, to our knowledge, none of these models lead exactly to a correction of the two-point function with the functional dependence of $q$ appearing in (59). It would be very interesting to understand this problem in more detail and to engineer a model that could reproduce our gravitational results.

One way we could engineer an SYK-like model that captures these features is to look at supersymmetric black holes in AdS$_5$. This would entail incorporating the presence of gauge fields in our five-dimensional Lagrangian, so that the black hole solutions can carry electromagnetic charges. This setup can accommodate, among others, for extremal supersymmetric rotating black holes in AdS$_5$ [60], for which a microstate counting procedure is available in the context of the AdS$_5$/CFT$_4$ correspondence. It would be interesting to study near-extremal deformations of the latter solutions in connection to nearly-AdS$_2$ holography.

## Acknowledgements

We are grateful to Jay Armas, Victor Godet, Joan Simón, Wei Song, Douglas Stanford, and Boyang Yu for discussions on this topic. The work of AC, CT, and EV is supported in part by the Delta ITP consortium, a program of the NWO that is funded by the Dutch Ministry of Education, Culture and Science (OCW). JFP is supported by the Simons Foundation through *It from Qubit: Simons Collaboration on Quantum Fields, Gravity, and Information*. CT acknowledges support from the NWO Physics/f grant n. 680-91-005. The work of EV is part of the research programme of the Foundation for Fundamental Research on Matter (FOM), which is financially supported by NWO.

## A  Aspects of the black hole background

In this appendix we collect some useful properties of the AdS$_5$ black hole that are used in the main sections. As mentioned in the main text, we focus on neutral 5D black holes with two

coincident angular momenta [31].

The mass and angular momentum can be obtained via holographic renormalization and Komar integral respectively, and read [36]

$$M = M_C + \frac{2\pi^2 m \left(3 + \frac{a^2}{\ell_5^2}\right)}{\kappa_5^2 \left(1 - \frac{a^2}{\ell_5^2}\right)^3} , \qquad J = \frac{8\pi^2 ma}{\kappa_5^2 \left(1 - \frac{a^2}{\ell_5^2}\right)^3} , \tag{139}$$

where $M_C$ is the Casimir energy for the case of equal angular momenta:

$$M_C = \frac{3\pi^2 \ell_5^2}{4\kappa_5^2} . \tag{140}$$

With reference to the metric (76), it is easy to see that the event horizon $\hat{r}_+$ is given by the largest root of $\Delta(\hat{r}_+) = 0$. It is most convenient to solve for $m$ instead of $\hat{r}_+$, which gives

$$m = \frac{\hat{r}_+^4 \left(\Xi + \frac{\hat{r}_+^2}{\ell_5^2}\right)}{2(\hat{r}_+^2 - a^2)} . \tag{141}$$

In this parametrization, the Bekenstein-Hawking entropy of the black hole is

$$S = \frac{4\pi^3 \hat{r}_+^4}{\kappa_5^2 \Xi^2 \sqrt{\hat{r}_+^2 - a^2}} , \tag{142}$$

and the temperature and rotational velocity respectively read

$$T = \frac{(\hat{r}_+^2 - 2a^2)\ell_5^2 + 2(\hat{r}_+^2 - a^2)^2}{2\pi \ell_5^2 \hat{r}_+^2 \sqrt{\hat{r}_+^2 - a^2}} , \qquad \Omega = \frac{a \left(\Xi + \frac{\hat{r}_+^2}{\ell_5^2}\right)}{\hat{r}_+^2} . \tag{143}$$

The thermodynamic quantities obey the first law of thermodynamics:

$$dM = TdS + \Omega dJ . \tag{144}$$

The Helmholtz free energy, i.e. the thermodynamic potential in the ensemble of fixed temperature and angular momentum, is

$$F(T,J) = M - TS , \tag{145}$$

while the Gibbs free energy

$$G(T,\Omega) = M - TS - \Omega J , \tag{146}$$

appears naturally as the (appropriately renormalized) Euclidean on-shell action $I_5 = \beta G$ [36].

Rotating black holes in AdS can suffer from a superradiant instability. Inside the so-called ergoregion – a region where no static observer is allowed – energy extraction becomes possible for some modes (with $\omega < m\Omega$). In [61], it was shown that rotating black holes in AdS$_5$ are stable against this phenomenon for $|\Omega|\ell_5 < 1$, where $\Omega$ is the rotational velocity:

$$\Omega = \frac{a(\Xi + \frac{\hat{r}_+^2}{\ell_5^2})}{\hat{r}_+^2} . \tag{147}$$

Rewriting the condition $\Omega\ell_5 < 1$ gives a condition on the horizon radius

$$\hat{r}_+^2 < a(\ell_5 + a) . \tag{148}$$

If this is satisfied, the black hole is stable against superradiance; conversely, violation of this condition is necessary for superradiance to occur.

## A.1 Near-extremality and decoupling limit

Extremality is achieved when $\hat{r}_+$ is a double zero of $\Delta(\hat{r})$, and as usual, the Hawking temperature (143) vanishes at extremality. Denoting the values of $(\hat{r}_+, a)$ at extremality by $(\hat{r}_0, a_0)$, extremality is imposed by taking

$$\ell_5^2 = \frac{2(a_0^2 - \hat{r}_0^2)^2}{2a_0^2 - \hat{r}_0^2} \, . \tag{149}$$

The extremal mass and angular momentum are

$$M_{\text{ext}} = M_C + \frac{2\pi^2}{\kappa_5^2} \frac{(a_0^2 - \hat{r}_0^2)^2 (8a_0^4 - 13a_0^2 \hat{r}_0^2 + 6\hat{r}_0^4)}{(2\hat{r}_0^2 - 3a_0^2)^3} \, , \tag{150}$$

and

$$J_{\text{ext}} = \frac{16\pi^2}{\kappa_5^2} \frac{a_0(a_0^2 - \hat{r}_0^2)^4}{(2\hat{r}_0^2 - 3a_0^2)^3} \, . \tag{151}$$

Notice that we can alternatively introduce variables $x = \sqrt{\frac{a_0^2}{\hat{r}_0^2 - a_0^2}}$ and $y = a/\ell_5$, as was done in [18]. Using these variables, the extremality condition (149) is equivalent to the relation $y^2 = \frac{1}{2}x^2(x^2 - 1)$.

In connection to the superradiant instability described above, for the extremal black hole we find

$$\Omega_{\text{ext}} \ell_5 < 1 \quad \Longleftrightarrow \quad \frac{2 - x^2}{2(x^2 - 1)} < 0 \, , \tag{152}$$

which is never true as $x \in (1, \sqrt{2})$. Thus, the extremal black holes considered here are not stable against superradiance.

The near-horizon region of the extremal rotating AdS$_5$ black hole is obtained as follows. We introduce a new radial coordinate $r$ given by

$$\hat{r} = \hat{r}_0 + \lambda r \, . \tag{153}$$

For the limit to be well-defined, we must also rescale

$$\hat{t} = \frac{\lambda_0}{\lambda} t \, ,$$
$$\hat{\psi} = \hat{\psi}_{\text{IR}} + \frac{\Omega_0}{\lambda} t \, , \tag{154}$$

where

$$\lambda_0^2 = \frac{\ell_5^4}{16} \frac{(x^2 - 1)^2 (x^2 + 1)}{(1 - 2x^2)^2} \, ,$$
$$\Omega_0 = \frac{\sqrt{2}\lambda_0}{\ell_5} \frac{x(2 - x^2)}{\sqrt{(x^2 - 1)}} \, . \tag{155}$$

We find the near-horizon geometry by expanding the five-dimensional metric (76) to leading order in $\lambda$; this gives

$$g_{\mu\nu}^{(5)} dx^\mu dx^\nu \rightarrow e^{\chi_0 + \psi_0} \ell_2^2 \left( -r^2 dt^2 + \frac{dr^2}{r^2} \right) + R^2 e^{-2\psi_0 + \chi_0} d\Omega_2^2$$
$$+ R^2 e^{-2\chi_0} \left( \sigma_{\text{IR}}^3 + \bar{A}_t dt \right)^2 + O(\lambda) \, , \tag{156}$$

where

$$\bar{A}_t = \frac{4\,\lambda_0}{\ell_5^2} \frac{(2-x^2)\,x}{(x^2-1)\sqrt{x^2+1}}\,r\,, \tag{157}$$

and

$$e^{2\chi_0} = \frac{2R^2}{\ell_5^2} \frac{(2-x^2)^2}{(x^2-1)}\,,$$
$$e^{2\psi_0} = \frac{2^{5/2}R^3}{\ell_5^3} \frac{(2-x^2)^2}{(x^2-1)^{3/2}}\,, \tag{158}$$

while the value of the two-dimensional cosmological constant is

$$\frac{1}{\ell_2^2} = \sqrt{\frac{8R^5 x^9}{a_0^9}(2-x^2)^2(2x^2-1)}\,. \tag{159}$$

These expressions are equivalent to those in (82)-(86) and to those in Sec. 3.1; here we have simply expressed them explicitly in terms of the parameters of the black hole. Note that we translated $(\hat{r}_0, a_0)$ to $x$, rather than $q$. The relation between them is

$$q = \frac{x^2-1}{4(2-x^2)}\,. \tag{160}$$

**Near-extremality.** It is possible to generalize the above discussion to the situation in which there is a small departure of $(\hat{r}_+, a)$ from $(\hat{r}_0, a_0)$. In particular, we will slightly increase the temperature above zero and the mass above (150), while keeping the angular momentum $J$ (and the AdS radius $\ell_5$) fixed. Near extremality the following relation between the mass $M$ and the temperature $T$ holds

$$M - M_{\text{ext}} = \frac{1}{M_{\text{gap}}}T^2 + O(T^3)\,, \tag{161}$$

where the mass gap $M_{gap}$ is a (dimensionful) quantity that quantifies the breaking of scaling symmetry. $M_{\text{gap}}$ is the scale of the smallest excitation energy of the black hole [3].[12] The mass gap shows up also in the low-temperature expansion of the entropy:

$$S = S_{\text{ext}} + \frac{2}{M_{\text{gap}}}T + O(T^2)\,, \tag{162}$$

as a consequence of the first law of thermodynamics. Defining the heat capacity at fixed angular momentum as

$$C_J \equiv T\left(\frac{dS}{dT}\right)_J\,, \tag{163}$$

the mass gap is given by

$$M_{\text{gap}} = \frac{2T}{C_J|_{T=0}} = \frac{\kappa_5^2}{2\pi^4\ell_5^4}\frac{(2-x^2)^2(2x^2-1)}{(3-x^2)(x^2-1)^2}\,. \tag{164}$$

A proper understanding of near-extremal black holes should account for a microscopic interpretation of this mass gap.

---

[12]This interpretation of $M_{\text{gap}}$ assumes that quantum corrections do not overwhelm the leading semi-classical correction. However, as pointed out recently in [20, 22], logarithmic corrections to the gravitational path integral can tamper with this interpretation and more care is needed to give these expressions an appropriate statistical interpretation.

We can also incorporate the near-extremal limit in the near-horizon geometry (156) by modifying the decoupling limit (153)-(154). In terms of the parameters of the black hole, we introduce a small departure from the extremal limit as

$$\hat{r}_+ = \hat{r}_0 + \varepsilon\lambda + O(\lambda^2)\,, \quad a = a_0 + O(\lambda^2)\,, \tag{165}$$

where $\lambda\varepsilon \ll \hat{r}_0$ and $\varepsilon$ is dimensionless. In this context, we are increasing the temperature of the black hole by

$$T = \frac{x^2(2x^2-1)}{\pi a_0^2\sqrt{1+x^2}}\varepsilon\lambda + O(\lambda^2)\,. \tag{166}$$

To reach the near-horizon region for the near-extremal black hole, the change of coordinates is

$$\hat{r} = \hat{r}_0 + \lambda\left(r + \frac{\varepsilon^2}{r}\right),$$
$$\hat{t} = \frac{\lambda_0}{\lambda}t\,, \tag{167}$$
$$\hat{\psi} = \hat{\psi}_{\mathrm{IR}} + \frac{\Omega_0}{\lambda}t\,,$$

for which, at leading order in $\lambda$, the metric reads

$$\begin{aligned} g^{(5)}_{\mu\nu}\mathrm{d}x^\mu\mathrm{d}x^\nu \to &-e^{\chi_0+\psi_0}\ell_2^2\frac{(r^2-\varepsilon^2)^2}{r^2}\mathrm{d}t^2 + e^{\chi_0+\psi_0}\ell_2^2\frac{\mathrm{d}r^2}{r^2}\\ &+ R^2 e^{-2\psi_0+\chi_0}d\Omega_2^2 + e^{-2\chi_0}\left(\sigma_{\mathrm{IR}}^3 + \bar{A}_t\mathrm{d}t\right)^2 + O(\lambda)\,, \end{aligned} \tag{168}$$

with

$$\bar{A}_t = \frac{4\lambda_0}{\ell_5^2}\frac{x\left(2-x^2\right)}{(x^2-1)\sqrt{x^2+1}}\left(r + \frac{\varepsilon^2}{r}\right)\,. \tag{169}$$

# B    Witten diagrams on AdS$_2$: how to deal with the irrelevant deformation

In Sec. 3.5, we considered the corrections to the two-point function of $\check{\mathcal{X}}$ as a result of the interaction terms between $\check{\mathcal{X}}$ and $\mathcal{Y}$ in the (cubic) Lagrangian (55). Subsequently, in Sec. 3.6, we discussed the corrections on the two- and four-point function as a result of the gravitational backreaction. In this appendix, we will first review some known AdS/CFT results on correlation functions, and then use these results to explicitly compute the corrections given in (59). We also give details on gravitational corrections and their effect on the OTOC.

## B.1    Tree level two- and three-point functions

For the calculation of correlation functions, we will need to compute a set of Witten diagrams on AdS$_2$. We note that several results have already been worked out in the literature, which we can directly apply to our problem [7, 62–64]. To be self-contained, in this appendix we will review the results of [63] for the tree level two- and three-point functions of general scalar operators. The calculations in that paper were carried out in the context of AdS$_{d+1}$/CFT$_d$, so in order to apply their results we can simply set $d = 1$.

In [63] the authors considered a generic Euclidean bulk action of the form

$$S_{\mathrm{E}} = \int d^{d+1}x\sqrt{g}\left[\frac{1}{2}(\nabla\phi_i)^2 + \frac{1}{2}m_i^2\phi_i^2 + \lambda_{ijk}\phi_i\phi_j\phi_k + \tilde{\lambda}_{ijk}\phi_i(\nabla\phi_j)(\nabla\phi_k)\right]. \tag{170}$$

Here $\phi_i$ ($i = 1, 2, 3, \ldots$) represent a set of minimally coupled bulk scalar fields with masses $m_i$, dual to a set of scalar operators $\mathcal{O}_i$ with conformal dimensions

$$\Delta_i = \frac{d + \sqrt{d^2 + 4m_i^2 \ell_{d+1}^2}}{2}, \tag{171}$$

and $\lambda_{ijk}$ and $\tilde{\lambda}_{ijk}$ are arbitrary cubic couplings. They showed that this system leads to the following (vacuum) two- and three-point functions for the dual operators:

$$\langle \mathcal{O}_i(\vec{x}_1) \mathcal{O}_j(\vec{x}_2) \rangle = \frac{\delta_{ij} D_i}{|\vec{x}_{12}|^{2\Delta_i}}, \tag{172}$$

and

$$\langle \mathcal{O}_i(\vec{x}_1) \mathcal{O}_j(\vec{x}_2) \mathcal{O}_k(\vec{x}_3) \rangle = \frac{\lambda_{ijk} K_{ijk} + \tilde{\lambda}_{ijk} \tilde{K}_{ijk}}{|\vec{x}_{12}|^{\Delta_i + \Delta_j - \Delta_k} |\vec{x}_{23}|^{\Delta_j + \Delta_k - \Delta_i} |\vec{x}_{31}|^{\Delta_k + \Delta_i - \Delta_j}}. \tag{173}$$

Here, $\vec{x}_{ij} \equiv \vec{x}_i - \vec{x}_j$, and we have

$$D_i = \frac{(2\Delta_i - 1)\Gamma[\Delta_i]}{\pi^{\frac{d}{2}} \Gamma[\Delta_i - \frac{d}{2}]}, \tag{174}$$

and

$$K_{ijk} = -\frac{\Gamma[\frac{1}{2}(\Delta_i + \Delta_j - \Delta_k)] \Gamma[\frac{1}{2}(\Delta_j + \Delta_k - \Delta_i)] \Gamma[\frac{1}{2}(\Delta_k + \Delta_i - \Delta_j)]}{2\pi^d \Gamma[\Delta_i - \frac{d}{2}] \Gamma[\Delta_j - \frac{d}{2}] \Gamma[\Delta_k - \frac{d}{2}]}$$
$$\times \Gamma[\frac{1}{2}(\Delta_i + \Delta_j + \Delta_k - d)], \tag{175}$$

$$\tilde{K}_{ijk} = \frac{K_{ijk}}{\ell_{d+1}^2} \left[ \Delta_j \Delta_k + \frac{1}{2}(d - \Delta_i - \Delta_j - \Delta_k)(\Delta_j + \Delta_k - \Delta_i) \right]. \tag{176}$$

As expected, $K_{ijk}$ is symmetric with respect to the three indices, but $\tilde{K}_{ijk}$ is only symmetric under $j \leftrightarrow k$.

The above correlators imply that the CFT effective action should contain the following terms:

$$
\begin{aligned}
I_{\text{eff}} = &-\delta_{ij} \frac{D_i}{2} \int \frac{d^d \vec{x}_1 d^d \vec{x}_2 \, \tilde{\phi}_i(\vec{x}_1) \tilde{\phi}_j(\vec{x}_2)}{|\vec{x}_{12}|^{2\Delta_i}} \\
&+ (\lambda_{ijk} K_{ijk} + \tilde{\lambda}_{ijk} \tilde{K}_{ijk}) \int \frac{d^d \vec{x}_1 d^d \vec{x}_2 d^d \vec{x}_3 \, \tilde{\phi}_i(\vec{x}_1) \tilde{\phi}_j(\vec{x}_2) \tilde{\phi}_k(\vec{x}_3)}{|\vec{x}_{12}|^{\Delta_i + \Delta_j - \Delta_k} |\vec{x}_{23}|^{\Delta_j + \Delta_k - \Delta_i} |\vec{x}_{31}|^{\Delta_k + \Delta_i - \Delta_j}} + \cdots,
\end{aligned} \tag{177}
$$

where $\tilde{\phi}_i$ are the sources and the couplings $\lambda_{ijk}$ and $\tilde{\lambda}_{ijk}$ include symmetry factors. Then

$$\langle \mathcal{O}_{i_1}(\vec{x}_1) \cdots \mathcal{O}_{i_n}(\vec{x}_n) \rangle = (-1)^{n+1} \left. \frac{\delta^n I_{\text{eff}}}{\delta \tilde{\phi}_{i_1}(\vec{x}_1) \cdots \delta \tilde{\phi}_{i_n}(\vec{x}_n)} \right|_{\tilde{\phi}_{i_k} = 0}. \tag{178}$$

It is important to note that all the above formulas are valid in Euclidean signature. In order to translate to real time, $x_0 \to it$, we can simply change $\sqrt{g} \to \sqrt{-g}$ in the action (170) and add an overall minus sign. In terms of the resulting correlators and the effective action, it suffices to change $|\vec{x}_{ij}| \to |(\vec{x}_i - \vec{x}_j)^2 - (t_i - t_j)^2|^{1/2}$, where $\vec{x}_i$ are now spatial vectors (with $(d-1)$ components). For $d = 1$ we can replace $|\vec{x}_{ij}| \to |t_i - t_j|$.

## B.2 Corrections to the two-point functions due to a background dilaton

We will now apply the results of the previous subsection to our problem. Recall that our bulk theory contains interaction terms between $\check{\mathcal{X}}$ and $\mathcal{Y}$ of the form

$$
\begin{aligned}
S_{\mathrm{E}}^{\mathrm{2D}} = \int d^2x \sqrt{g} \Bigg[ &\frac{1}{2}(\nabla\check{\mathcal{X}})^2 + \frac{1}{2}m_{\mathcal{X}}^2 \check{\mathcal{X}}^2 + \lambda_{\mathcal{Y}\check{\mathcal{X}}\check{\mathcal{X}}} \mathcal{Y}\check{\mathcal{X}}^2 \\
&+ \tilde{\lambda}_{\check{\mathcal{X}}(\partial\check{\mathcal{X}})(\partial\mathcal{Y})} \check{\mathcal{X}}(\nabla\check{\mathcal{X}})(\nabla\mathcal{Y}) + \tilde{\lambda}_{\mathcal{Y}(\partial\check{\mathcal{X}})(\partial\check{\mathcal{X}})} \mathcal{Y}(\nabla\check{\mathcal{X}})^2 \Bigg],
\end{aligned}
\tag{179}
$$

which are exactly as in (170). For our particular system, we have

$$
m_{\mathcal{X}}^2 \equiv \frac{1}{\ell_2^2} \frac{6 + 32q}{1 + 12q},
\tag{180}
$$

and the coupling constants are

$$
\lambda_{\mathcal{Y}\check{\mathcal{X}}\check{\mathcal{X}}} = -\frac{e^{2\psi_0}(1+6q)(9+38q-80q^2)}{2\ell_2^2(1+2q)^2(1+12q)},
$$
$$
\tilde{\lambda}_{\check{\mathcal{X}}(\partial\check{\mathcal{X}})(\partial\mathcal{Y})} = \frac{12q^2 e^{2\psi_0}}{(1+2q)^2}, \qquad \tilde{\lambda}_{\mathcal{Y}(\partial\check{\mathcal{X}})(\partial\check{\mathcal{X}})} = \frac{e^{2\psi_0}}{2}.
\tag{181}
$$

However, in this appendix we will take these constants to be arbitrary for the sake of generality. The results below will therefore generalize the appendix C of [7] to more general couplings between the dilaton and matter; see also [65].

The idea is to treat $\mathcal{Y}$ as a background field and study the effect on the two-point function of $\check{\mathcal{X}}$. This yields the leading order correction above the free result, as depicted in the second diagram of Fig. 1. Using the results of the previous subsection (specializing to $d = 1$), we can write several terms for the 1D effective action in the vacuum:

$$
I_{\mathrm{eff}} = I_{\mathrm{free}} + I_{\mathrm{interactions}}.
\tag{182}
$$

The free part yields

$$
I_{\mathrm{free}} = -\frac{D}{2} \int dt_1 dt_2 \frac{\check{\mathcal{X}}(t_1)\check{\mathcal{X}}(t_2)}{|t_1 - t_2|^{2\Delta}},
\tag{183}
$$

where

$$
D = \frac{(2\Delta - 1)\Gamma[\Delta]}{\sqrt{\pi}\Gamma[\Delta - \frac{1}{2}]}, \qquad \Delta \equiv \Delta_{\mathcal{X}} = \frac{1 + \sqrt{1 + 4m_{\mathcal{X}}^2\ell_2^2}}{2}.
\tag{184}
$$

Similarly, we can write down three interactions terms corresponding to the terms proportional to $\mathcal{Y}\check{\mathcal{X}}^2$, $\check{\mathcal{X}}(\nabla\check{\mathcal{X}})(\nabla\mathcal{Y})$ and $\mathcal{Y}(\nabla\check{\mathcal{X}})^2$ in (179). Here, we can assume that $\mathcal{Y}$ corresponds to an operator of dimension $\Delta_{\mathcal{Y}} = -1$ [7]. In addition, we need specific coefficients of the type (175)-(176):

$$
K_{\mathcal{Y}\check{\mathcal{X}}\check{\mathcal{X}}} = -\frac{\Gamma[-\frac{1}{2}]^2 \Gamma[\Delta + \frac{1}{2}]\Gamma[\Delta - 1]}{2\pi\Gamma[\Delta - \frac{1}{2}]^2 \Gamma[-\frac{3}{2}]} = -\frac{3(\Delta - \frac{1}{2})\Gamma[\Delta - 1]}{2\sqrt{\pi}\Gamma[\Delta - \frac{1}{2}]},
\tag{185}
$$

$$
\tilde{K}_{\check{\mathcal{X}}(\partial\check{\mathcal{X}})(\partial\mathcal{Y})} = \frac{K_{\mathcal{Y}\check{\mathcal{X}}\check{\mathcal{X}}}}{\ell_2^2}\left[-\Delta + \frac{1}{2}(2-2\Delta)(-1)\right] = -\frac{K_{\mathcal{Y}\check{\mathcal{X}}\check{\mathcal{X}}}}{\ell_2^2},
\tag{186}
$$

and

$$
\tilde{K}_{\mathcal{Y}(\partial\check{\mathcal{X}})(\partial\check{\mathcal{X}})} = \frac{K_{\mathcal{Y}\check{\mathcal{X}}\check{\mathcal{X}}}}{\ell_2^2}\left[\Delta^2 + \frac{1}{2}(2-2\Delta)(2\Delta+1)\right] = -(\Delta^2 - \Delta - 1)\frac{K_{\mathcal{Y}\check{\mathcal{X}}\check{\mathcal{X}}}}{\ell_2^2}.
\tag{187}
$$

The sum of these terms leads to the following interaction term in the effective action:

$$I_{\text{interactions}} = \frac{\tilde{D}}{2} \int \frac{dt_1 dt_2 dt_3 \, \breve{\mathcal{X}}(t_1)\breve{\mathcal{X}}(t_2)\breve{\mathcal{Y}}(t_3)}{|t_{12}|^{2\Delta+1}|t_{23}|^{-1}|t_{31}|^{-1}},$$

(188)

where

$$\tilde{D} \equiv \lambda_{\mathcal{Y}\breve{\mathcal{X}}\breve{\mathcal{X}}} K_{\mathcal{Y}\breve{\mathcal{X}}\breve{\mathcal{X}}} + \tilde{\lambda}_{\breve{\mathcal{X}}(\partial\breve{\mathcal{X}})(\partial\mathcal{Y})} \tilde{K}_{\breve{\mathcal{X}}(\partial\breve{\mathcal{X}})(\partial\mathcal{Y})} + \tilde{\lambda}_{\mathcal{Y}(\partial\breve{\mathcal{X}})(\partial\breve{\mathcal{X}})} \tilde{K}_{\mathcal{Y}(\partial\breve{\mathcal{X}})(\partial\breve{\mathcal{X}})}.$$

(189)

Let us now compute the two-point function (in a thermal state), and see how the interaction terms correct the free result. In the vacuum (pure AdS in the bulk), we can use coordinates such that

$$ds^2_{\text{AdS}_2} = \frac{\ell_2^2}{z^2}(dt^2 + dz^2).$$

(190)

Note that the boundary is at $z \to 0$, whereas in section 3 we used Poincaré coordinates $(t, r)$ with the boundary located at $r \to \infty$. In these coordinates, the near-boundary expansion of $\breve{\mathcal{X}}$ is such that

$$\breve{\mathcal{X}}(t, z) = z^{1-\Delta}\tilde{\mathcal{X}}(t) + \cdots, \qquad \text{as } z \to 0,$$

(191)

and $\tilde{\mathcal{X}}(t)$ is interpreted as the source of $\mathcal{O}_{\breve{\mathcal{X}}}$. Next, we can perform a diffeomorphism in the bulk to go to a thermal state. The important point in this transformation is to keep track of the UV cutoff, which follows a general trajectory $\{t(u), z(u)\}$ ($u$ here can be interpreted as a boundary time), with

$$g|_{\text{bndy}} = \frac{\ell_2^2}{\epsilon^2} = \text{const.}$$

(192)

The above condition implies that

$$z = \epsilon\sqrt{(t')^2 + (z')^2} = \epsilon t' + \mathcal{O}(\epsilon^3).$$

(193)

Under this transformation, the asymptotic form of the field $\breve{\mathcal{X}}$ becomes

$$\breve{\mathcal{X}}(t, z) = \epsilon^{1-\Delta}[t'(u)]^{1-\Delta}\tilde{\mathcal{X}}(t(u)) + \cdots, \qquad \text{as } \epsilon \to 0,$$

(194)

and now $[t'(u)]^{1-\Delta}\tilde{\mathcal{X}}(t(u)) \equiv \bar{\mathcal{X}}(u)$ is interpreted as a source. Hence, the different terms in the effective action transform accordingly; in particular, the free part now reads:

$$I_{\text{free}} = -\frac{D}{2} \int du_1 du_2 \left[\frac{t'(u_1)t'(u_2)}{|t(u_1) - t(u_2)|^2}\right]^{\Delta} \bar{\mathcal{X}}(u_1)\bar{\mathcal{X}}(u_2).$$

(195)

For a thermal state, we have

$$t(u) = \tan\left(\frac{\frac{2\pi}{\beta}u}{2}\right), \qquad u \sim u + \beta,$$

(196)

and hence

$$\langle \mathcal{O}_{\mathcal{X}}(u_1)\mathcal{O}_{\mathcal{X}}(u_2)\rangle^{\text{free}}_{\beta} = D\left[\frac{t'(u_1)t'(u_2)}{|t(u_1) - t(u_2)|^2}\right]^{\Delta} = D\left[\frac{\pi}{\beta \sin(\frac{\pi u_{12}}{\beta})}\right]^{2\Delta},$$

(197)

where $u_{12} \equiv u_1 - u_2$.

Equation (197) is the free result and gets corrected by the interaction terms. For the terms considered above in (188) we have, changing integration variables $t_i \to u_i$ to go to the thermal state,

$$I_{\text{interactions}} = \frac{\tilde{D}}{2} \int du_1 du_2 du_3 \frac{t'(u_1)^{\Delta}t'(u_2)^{\Delta}t'(u_3)^{-1}\bar{\mathcal{X}}(u_1)\bar{\mathcal{X}}(u_2)\bar{\mathcal{Y}}(u_3)}{|t(u_1) - t(u_2)|^{2\Delta+1}|t(u_1) - t(u_3)|^{-1}|t(u_2) - t(u_3)|^{-1}}.$$

(198)

We assume that the source is constant in the thermal frame, i.e. $\bar{\mathcal{Y}}(u_3) = a$. Then, we use (196) to obtain

$$
\begin{aligned}
\langle \mathcal{O}_{\mathcal{X}}(u_1)\mathcal{O}_{\mathcal{X}}(u_2)\mathcal{O}_{-1}(u_3)\rangle_\beta &= \tilde{D}\left[\frac{t'(u_1)t'(u_2)}{|t(u_1)-t(u_2)|^2}\right]^\Delta \frac{|t(u_1)-t(u_3)||t(u_2)-t(u_3)|}{t'(u_3)\,|t(u_1)-t(u_2)|} \\
&= \frac{\tilde{D}\beta}{\pi}\left[\frac{\pi}{\beta\sin(\frac{\pi u_{12}}{\beta})}\right]^{2\Delta} \frac{|\sin(\frac{\pi u_{13}}{\beta})|\,|\sin(\frac{\pi u_{23}}{\beta})|}{|\sin(\frac{\pi u_{12}}{\beta})|}.
\end{aligned}
\tag{199}
$$

Finally, integrating over $u_3$ we get the correction to the two-point function from these interactions. In order to do the integral we must be careful with absolute values. Assuming $u_1 > u_2$ we obtain

$$
\begin{aligned}
\int_0^\beta du_3 |\sin(\tfrac{\pi u_{13}}{\beta})|\,|\sin(\tfrac{\pi u_{23}}{\beta})| &= \left(\int_0^{u_2} du_3 - \int_{u_2}^{u_1} du_3 + \int_{u_1}^\beta du_3\right)\sin(\tfrac{\pi u_{13}}{\beta})\sin(\tfrac{\pi u_{23}}{\beta}) \\
&= \frac{\beta}{2\pi}\left[2\sin(\tfrac{\pi u_{12}}{\beta}) + \pi(1-\tfrac{2u_{12}}{\beta})\cos(\tfrac{\pi u_{12}}{\beta})\right],
\end{aligned}
\tag{200}
$$

so:

$$
\begin{aligned}
\langle \mathcal{O}_{\mathcal{X}}(u_1)\mathcal{O}_{\mathcal{X}}(u_2)\rangle_\beta^{\text{correction}} &= a\int_0^\beta du_3\,\langle \mathcal{O}_{\mathcal{X}}(u_1)\mathcal{O}_{\mathcal{X}}(u_2)\mathcal{O}_{-1}(u_3)\rangle_\beta \\
&= \frac{\tilde{D}a\beta^2}{2\pi^2}\left[\frac{\pi}{\beta\sin(\frac{\pi u_{12}}{\beta})}\right]^{2\Delta}\left(2 + \pi\frac{1-2u_{12}/\beta}{\tan(\frac{\pi u_{12}}{\beta})}\right).
\end{aligned}
\tag{201}
$$

Adding the free part of the correlator, we find

$$
\langle \mathcal{O}_{\mathcal{X}}(u_1)\mathcal{O}_{\mathcal{X}}(u_2)\rangle_\beta = \left[\frac{\pi}{\beta\sin(\frac{\pi u_{12}}{\beta})}\right]^{2\Delta}\left[D + \frac{\tilde{D}a\beta^2}{2\pi^2}\left(2 + \pi\frac{1-2u_{12}/\beta}{\tan(\frac{\pi u_{12}}{\beta})}\right)\right].
\tag{202}
$$

The constants $D$ and $\tilde{D}$ are given in (184) and (189), respectively. We point out that, while $D$ is positive definite, $\tilde{D}$ can have either sign, depending on the cubic couplings appearing in the action (179). We note that $K_{\mathcal{Y}\check{\mathcal{X}}\check{\mathcal{X}}} < 0$ and $\tilde{K}_{\check{\mathcal{X}}(\partial\check{\mathcal{X}})(\partial\mathcal{Y})} > 0$ for $\Delta \geq 1$, while $\tilde{K}_{\mathcal{Y}(\partial\check{\mathcal{X}})(\partial\check{\mathcal{X}})} < 0$ in the range $1 \leq \Delta \leq \frac{1}{2}(1+\sqrt{5}) \approx 1.618$, or $\tilde{K}_{\mathcal{Y}(\partial\check{\mathcal{X}})(\partial\check{\mathcal{X}})} > 0$ otherwise. Putting all together, we conclude that $\tilde{D} < 0$ whenever the following condition is satisfied:

$$
\tilde{\lambda}_{\check{\mathcal{X}}(\partial\check{\mathcal{X}})(\partial\mathcal{Y})} + \left(\Delta^2 - \Delta - 1\right)\tilde{\lambda}_{\mathcal{Y}(\partial\check{\mathcal{X}})(\partial\check{\mathcal{X}})} < \ell_2^2\lambda_{\mathcal{Y}\check{\mathcal{X}}\check{\mathcal{X}}}.
\tag{203}
$$

### B.3 Gravitational effects on the two- and four-point functions

The leading correction to the two-point function at order $\mathcal{O}(G_N)$ is given by the third diagram in Fig. 1. As explained in Sec. 3, we can do the calculation directly from this diagram or, alternatively, we can calculate it from an appropriate diffeomorphism. We will follow the latter approach, which is simpler.

In the calculation in App. B.2 we used a diffeomorphism $t(u)$, given in (196), which gives the saddle point of the Schwarzian theory; if we were to consider gravitational loop corrections to the matter correlators coming from the Schwarzian mode we can simply expand around the thermal saddle[13]

$$
t(u) = \tan\left(\frac{u+\varepsilon(u)}{2}\right),
\tag{204}
$$

---

[13]For the ease of notation we normalize the temperature to $\beta = 2\pi$, but it can be restored later on by dimensional analysis.

and use the $n$-point functions $\langle \varepsilon(u_1) \cdots \varepsilon(u_n) \rangle$ computed from the Schwarzian theory. We assume that the latter is normalized such that the effective action includes a term of the form [7,34]

$$I_{\text{grav}} = -C \int du \, \{t, u\}, \tag{205}$$

where

$$\{t, u\} \equiv -\frac{1}{2} \frac{t''^2}{t'^2} + \left( \frac{t''}{t'} \right)' = \frac{t'''}{t'} - \frac{3}{2} \left( \frac{t''}{t'} \right)^2 . \tag{206}$$

From equation (6.41) of [18] we then know that in our system

$$C = \frac{\ell_2^2 a}{\kappa_2^2} = \frac{\ell_2^2 a}{8 \pi G_N} . \tag{207}$$

For practical purposes (one-loop calculations) all we need to know are the one- and two-point functions [7,34]:

$$\langle \varepsilon(u) \rangle = 0 ,$$
$$\langle \varepsilon(u)\varepsilon(0) \rangle \equiv G(u) = \frac{2\pi}{C} \left[ -\frac{(u-\pi)^2}{2} + (u-\pi)\sin u + 1 + \frac{\pi^2}{6} + \frac{5}{2}\cos u \right], \tag{208}$$

together with the relations

$$\begin{aligned}
\langle \varepsilon'(u) \rangle &= 0 , \\
\langle \varepsilon(u_1)\varepsilon(u_2) \rangle &= G(|u_{12}|) , \\
\langle \varepsilon'(u_1)\varepsilon(u_2) \rangle &= \operatorname{sgn} u_{12} G'(|u_{12}|) , \\
\langle \varepsilon'(u_1)\varepsilon'(u_2) \rangle &= -G''(|u_{12}|) .
\end{aligned} \tag{209}$$

The leading one loop correction of the two-point function can be obtained by plugging (204) into the free action (195) and expanding up to quadratic order in $\varepsilon$. As a result one finds

$$I_{\text{matter}} = -\frac{D}{2} \int \frac{du_1 du_2}{\left[ 2\sin(\frac{u_{12}}{2}) \right]^{2\Delta}} \left[ 1 + \langle \mathcal{B}(u_1, u_2) \rangle + \langle \mathcal{C}(u_1, u_2) \rangle + \mathcal{O}(\varepsilon^3) \right] \bar{\mathcal{X}}(u_1)\bar{\mathcal{X}}(u_2), \tag{210}$$

where

$$\begin{aligned}
\mathcal{B}(u_1, u_2) &= \Delta \left( \varepsilon'(u_1) + \varepsilon'(u_2) - \frac{\varepsilon(u_1) - \varepsilon(u_2)}{\tan \frac{u_{12}}{2}} \right), \\
\mathcal{C}(u_1, u_2) &= \frac{\Delta}{\left( 2\sin\frac{u_{12}}{2} \right)^2} \Big[ (1 + \Delta + \Delta \cos u_{12})(\varepsilon(u_1) - \varepsilon(u_2))^2 \\
&\quad + 2\Delta \sin u_{12}(\varepsilon(u_2) - \varepsilon(u_1))(\varepsilon'(u_1) + \varepsilon'(u_2)) \\
&\quad - (\cos u_{12} - 1)\big( (\Delta - 1)(\varepsilon'(u_1)^2 + \varepsilon'(u_2)^2) + 2\Delta \varepsilon'(u_1)\varepsilon'(u_2) \big) \Big] .
\end{aligned} \tag{211}$$

Using (208)-(209) one can then check that

$$\begin{aligned}
\langle \mathcal{B}(u_1, u_2) \rangle &= 0 , \\
\langle \mathcal{C}(u_1, u_2) \rangle &= \frac{1}{2\pi C} \frac{\Delta}{\left( 2\sin\frac{u_{12}}{2} \right)^2} \Big[ 2 + 4\Delta + u_{12}(u_{12} - 2\pi)(\Delta + 1) \\
&\quad + \big( \Delta u_{12}(u_{12} - 2\pi) - 4\Delta - 2 \big) \cos u_{12} + 2(\pi - u_{12})(2\Delta + 1)\sin u_{12} \Big] ,
\end{aligned} \tag{212}$$

and therefore

$$\langle \mathcal{O}_\mathcal{X}(u_1)\mathcal{O}_\mathcal{X}(u_2)\rangle_\beta = \frac{D}{\left[2\sin(\frac{u_{12}}{2})\right]^{2\Delta}}\left[1 + \langle \mathcal{C}(u_1,u_2)\rangle + \mathcal{O}(\varepsilon^3)\right].\tag{213}$$

The "1" in the bracket gives the free result, while the term proportional to $\langle \mathcal{C}(u_1,u_2)\rangle$ is the contribution from the third diagram in Fig. 1 which, as explained there, is of order $\mathcal{O}(1/C) \sim \mathcal{O}(G_N/a)$.

The leading order result for the four-point function is of order $\mathcal{O}(G_N)$ and is given by the first diagram in Fig. 2. This is the graviton exchange diagram. To compute this diagram we also proceed by implementing a suitable diffeomorphism. Exponentiating (210) leads to the generator of connected correlators, which has the following terms

$$\begin{aligned}
\log\langle e^{-I_{\text{matter}}}\rangle = {} & \frac{D}{2}\int \frac{\mathrm{d}u_1\mathrm{d}u_2}{\left[2\sin(\frac{u_{12}}{2})\right]^{2\Delta}}\left[1 + \langle \mathcal{C}(u_1,u_2)\rangle\right]\bar{\mathcal{X}}(u_1)\bar{\mathcal{X}}(u_2)\\
& + \frac{D^2}{8}\int \frac{\mathrm{d}u_1\mathrm{d}u_2\mathrm{d}u_3\mathrm{d}u_4}{\left[4\sin(\frac{u_{12}}{2})\sin(\frac{u_{34}}{2})\right]^{2\Delta}}\langle \mathcal{B}(u_1,u_2)\mathcal{B}(u_3,u_4)\rangle\bar{\mathcal{X}}(u_1)\bar{\mathcal{X}}(u_2)\bar{\mathcal{X}}(u_3)\bar{\mathcal{X}}(u_4)\\
& + \mathcal{O}(G_N^2).
\end{aligned}\tag{214}$$

The second term here gives the leading diagram of the four-point function,

$$\langle \mathcal{O}_\mathcal{X}(u_1)\mathcal{O}_\mathcal{X}(u_2)\mathcal{O}_\mathcal{X}(u_3)\mathcal{O}_\mathcal{X}(u_4)\rangle_\beta \sim D^2\frac{\langle \mathcal{B}(u_1,u_2)\mathcal{B}(u_3,u_4)\rangle}{\left[2\sin(\frac{u_{12}}{2})\right]^{2\Delta}\left[2\sin(\frac{u_{34}}{2})\right]^{2\Delta}}.\tag{215}$$

Again, we can use (208)-(209) to evaluate the expectation value on the RHS. However, we have to be careful with time ordering, as the four-point function that is relevant to chaos is an OTOC. Here, one can proceed in Euclidean signature and then analytically continue to real time following the prescription of [35], i.e. setting $u_1 = i\epsilon_1$, $u_2 = t + i\epsilon_2$, $u_3 = i\epsilon_3$, $u_4 = t + i\epsilon_4$, with $\epsilon_1 = \beta/2$, $\epsilon_2 = -\beta/4$, $\epsilon_3 = 0$ and $\epsilon_4 = \beta/4$. This corresponds to the insertion of the operators at equal spacing around the thermal circle. At the end of the calculation one finds that the late time behavior of the OTOC is [7,34]

$$\langle \mathcal{B}(0,t)\mathcal{B}(0,t)\rangle \sim \frac{\beta\Delta^2}{C}e^{\frac{2\pi}{\beta}t}, \qquad (t \gg \beta),\tag{216}$$

with a Lyapunov exponent that saturates the chaos bound,

$$\lambda_L = \frac{2\pi}{\beta}.\tag{217}$$

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
