# Peer review of "Rotating 5D Black Holes: Interactions and deformations near extremality"

_SciPost Physics, doi:SciPost Phys. 11, 102 (2021)_

## Round 2 · Referee Report · Anonymous · 2021-8-18

Report
This is, at bottom, an analysis of certain gravitational perturbations of an extremal rotating black hole in five dimensions. They are cast in the framework of a KK reduction down to the "(t,r)" sector of the metric in the throat region. Attention is paid to the matching to the region outside the throat, and to the interpretation in terms of JT gravity and dual SYK models.
The gravitational perturbations of rotating black holes are notoriously thorny. In the present case, even with the simplifications of near-extremality and near-throat analysis, much of the complication remains, so the study is technically involved. As far as I've been able to see, without immersing myself fully in the calculational details, it looks thorough and correct. Unsurprisingly, the results aren't very neat, but an attempt has been made to extract in a meaningful way the main lessons learned, with an emphasis on the consequences of having or not a negative cosmological constant.
I find that the article is a useful addition to the literature, and it is overall well presented, so I am happy to recommend that it be published. I would like that the authors consider the following suggestions for improving its clarity, and also a possible extension. If the authors revise their article, I should not need to review the new version.
Requested changes
Suggested changes:
1-To prevent confusion early on: in the abstract, "black holes with a single rotational parameter in five dimensions" may mislead readers into thinking that black holes with rotation along a single plane are considered -- which they are not. It would be helpful to mention in the abstract and intro that the bhs considered have what is commonly referred as "equal angular momenta" (as indeed they are described in the appendix).
2-To guide the reader early on: I would have found quite helpful if in the context of the metrics (2.2) a brief discussion had been made of how the MP solution (for simplicity of illustration, without a cosmological constant) fits in this framework -- otherwise the discussion remains very abstract.
A suggested extension:
3-Besides the asymptotically flat and AdS set ups, the 5D Myers-Perry black holes are also present in the context of dyonic KK black holes, where they appear at the tip of a KK monopole geometry (as far as I'm aware, this was made explicit in hep-th/9809063 and in hep-th/0701150). Therefore it would seem natural, and possibly not too difficult, to extend the analysis of the present article to include the throat of these black holes near extremality. Given the interesting properties of these dyonic black holes, this would seem to be a worthwhile study.
Author: Chiara Toldo on 2021-10-15 [id 1854]
(in reply to Report 1 on 2021-08-18)
First of all, we would like to thank the referee for their useful comments and remarks on our manuscript.
Regarding the revision of the paper, we have done the following modifications:
1) we have replaced the descriptor "with a single rotational parameter" with "with equal angular momenta", as requested.
2) we have added the example of the Myers Perry solution in section 2.2, where we begin describing the procedure of dimensional reduction, as requested.
3) We thank the referee for mentioning KK BHs. The KK black hole in hep-th/9809063 is exactly the 5D Myers-Perry solution we are studying, and we have added a remark in section 2.2 to make this explicit. The solutions in hep-th/0701150 (which is based on solutions constructed in hep-th/0002166) are much more complicated to study and, while interesting per se, are outside our scope. We suggested instead to consider the solutions in 1009.5039 as a possible extension, which is now mentioned on page 27.
Author: Chiara Toldo on 2021-10-19 [id 1864]
(in reply to Report 2 on 2021-10-02)First of all, we would like to thank the referee for their useful comments and remarks on our manuscript.
In regard to the three comments in the report:
1) We added a remark at the end of section 3.5 (after eqn (3.48)) to stress that the background that has $\tilde{D}=0$ requires further attention. We are indeed very puzzled about it, and explored various possibilities in section 5, but it remains an open problem to understand why the value of $q = 2.85$ is special.
2)-3) We added a paragraph in section 5, page 27, to highlight other black holes for which our methods and setup can easily be extended. This includes other rotating black holes that would be as tractable as ours, and solutions in de Sitter. Some of these points are currently under investigation and we hope to report on them, in separate papers, soon. We also remark that adding rotation in 4D or considering a black ring configuration can be very cumbersome due to the intricate way that the rotational parameters enter in the geometry. This makes it more difficult to capture dynamical aspects of near-AdS$_2$, and we remark how one could perhaps overcome them.

---

## Round 2 · Referee Report · Anonymous · 2021-10-2

Report
This paper studies rotating 5d AdS black holes near extremality, with a view to understanding the JT gravity limit alongwith additional interaction terms induced by the single rotation parameter here. Many of the features here (e.g. corrections to correlation functions) are detail-dependent and give insight into the interplay between the IR JT sector and the embedding UV sector which is the full black hole environment.
The reduction ansatz eq.2.2 leads eventually to the effective 2d theory eq.2.9, building on the previous work 1807.06988 (schematically similar actions $\int (\phi.R-\phi.(\partial\Psi)^2-U(\phi,\Psi))$ have appeared previously in discussions of JT gravity, e.g. from nonrelativistic theories , but the present context is perhaps more intricate). The cubic perturbations around the JT sector can be organized as eqs.3.28-3.30. The scalar 2-pt function with cubic corrections stemming from the background dilaton is analysed in detail using well-known AdS/CFT technology for correlation functions applied to AdS2 (App.B.2) giving eq.3.45 (noted first in ref.[7] for a generic cubic interaction). The authors then map these to perturbations in the UV theory, noting specific differences between asymptotically AdS and flat cases.
The paper is interesting and contains many detailed investigations: I recommend publication. I give below a few comments/questions that the authors may consider, perhaps mostly towards future directions:
(1) Eq.3.45 is detail-dependent: ${\tilde D}$ changes sign depending on the q value defining the extremal point (occurring only for AdS black holes). The range found ensures that the scalar mode {\tilde\chi} is more irrelevant than the dilaton. The authors give some comments on the interpretation in sec.5, mostly ruling out various possibilities as far as I can see (including AdS black ring directions). It may be interesting to understand better the sign-changing point, i.e. the q-value where ${\tilde D}=0$.
(2) While the paper focusses on 5d black holes, I think the general setup should work for 4d near extremal rotating black holes as well (altho the details will differ) -- the authors may consider adding brief comments on this, which may be useful for completeness. Perhaps more comments on AdS black rings may also be useful, as relevant.
(3) Apart from a comment in Footnote-2, most of the paper pertains to AdS black holes or flat space. The studies in this paper may possibly be used in the context of dS2 JT gravity embedded in the near-Nariai limit of Schwarzschild dS4 black holes. The interplay between the IR JT and UV Schw dS were studied in arXiv:1904.01911 and other work. One may consider the effects of a small rotation parameter on the ndS2 limit by looking at near-Nariai Schw dS with a small rotation added -- then perhaps some of the technology here may be of value to understand parts of the space of ndS2 deformations (perhaps carefully considering appropriate analytic continuations).

---

## Round 3 · Author Response

First of all, we would like to thank the referees for their useful comments and remarks on our manuscript, especially those regarding possible further directions. Following their suggestions, we have elucidated some points in the text, and made minor revisions to the manuscript.

---

## Round 3 · List of Changes

We have done the following modifications:

1) We have replaced the descriptor ``with a single rotational parameter'' with ``with equal angular momenta,'' as requested.

2) We have added the example of the Myers Perry solution in section 2.2, where we begin describing the procedure of dimensional reduction, as requested.

3) We thank the referee for mentioning KK BHs. The KK black hole in hep-th/9809063 is exactly the 5D Myers-Perry solution we are studying, and we have added a remark in section 2.2 to make this explicit. The solutions in hep-th/0701150 (which is based on solutions constructed in hep-th/0002166) are much more complicated to study and, while interesting per se, are outside our scope. We suggested instead to consider the solutions in 1009.5039 as a possible extension, which is now mentioned on page 27.

4) We added a remark at the end of section 3.5 (after eqn (3.48)) to stress that the background that has $\tilde{D}=0$ requires further attention. We are indeed very puzzled about it, and explored various possibilities in section 5, but it remains an open problem to understand why the value of $q = 2.85$ is special.

5) We added a paragraph in section 5, page 27, to highlight other black holes for which our methods and setup can easily be extended. This includes other rotating black holes that would be as tractable as ours, and solutions in de Sitter. Some of these points are currently under investigation and we hope to report on them, in separate papers, soon. We also remark that adding rotation in 4D or considering a black ring configuration can be very cumbersome due to the intricate way that the rotational parameters enter in the geometry. This makes it more difficult to capture dynamical aspects of near-AdS$_2$, and we remark how one could perhaps overcome them.

---

## Editorial Decision

published